# From Canonical Correlation Analysis to Self-supervised Graph Neural Networks

**Hengrui Zhang**[1]*, **Qitian Wu**[2], **Junchi Yan**[2], **David Wipf**[3], **Philip S. Yu**[1]†
[1] Department of Computer Science, University of Illinois at Chicago
[2] Department of Computer Science and Engineering, Shanghai Jiao Tong University
[3] AWS Shanghai AI Lab
hzhan55@uic.edu, {echo740, yanjunchi}@sjtu.edu.cn
daviwipf@amazon.com, psyu@uic.edu

## Abstract

We introduce a conceptually simple yet effective model for self-supervised representation learning with graph data. It follows the previous methods that generate two views of an input graph through data augmentation. However, unlike contrastive methods that focus on instance-level discrimination, we optimize an innovative feature-level objective inspired by classical Canonical Correlation Analysis. Compared with other works, our approach requires none of the parameterized mutual information estimator, additional projector, asymmetric structures, and most importantly, negative samples which can be costly. We show that the new objective essentially 1) aims at discarding augmentation-variant information by learning invariant representations, and 2) can prevent degenerated solutions by decorrelating features in different dimensions. Our theoretical analysis further provides an understanding for the new objective which can be equivalently seen as an instantiation of the Information Bottleneck Principle under the self-supervised setting. Despite its simplicity, our method performs competitively on seven public graph datasets. The code is available at: https://github.com/hengruizhang98/CCA-SSG.

## 1   Introduction

Self-supervised learning (SSL) has been a promising paradigm for learning useful representations without costly labels [7, 46, 5]. In general, it learns representations via a proxy objective between inputs and self-defined signals, among which contrastive methods [46, 40, 16, 5, 12] have achieved impressive performance on learning image representations by maximizing the mutual information of two views (or augmentations) of the same input. Such methods can be interpreted as a discrimination of a joint distribution (positive pairs) from the product of two marginal ones (negative pairs) [50].

Inspired by the success of contrastive learning in vision [17, 46, 40, 5, 16, 12, 6], similar methods have been adapted to learning graph neural networks [48, 15, 33, 57, 58]. Although these models have achieved impressive performance, they require complex designs and architectures. For example, DGI [48] and MVGRL [15] rely on a parameterized mutual information estimator to discriminate positive node-graph pairs from negative ones; GRACE [57] and GCA [58] harness an additional MLP-projector to guarantee sufficient capacity. Moreover, negative pairs sampled or constructed from data often play an indispensable role in providing effective contrastive signals and have a large impact on performance. Selecting proper negative samples is often nontrivial for graph-structured data, not to mention the extra storage cost for prohibitively large graphs. BGRL [39] is a recent endeavor on

---

*This work was done during the author's internship at AWS Shanghai AI Lab.
†Corresponding author.

35th Conference on Neural Information Processing Systems (NeurIPS 2021).

Table 1: Technical comparison of self-supervised node representation learning methods. We provide a conceptual comparison with more self-supervised methods in Appendix G. *Target* denotes the comparison pair, N/G/F denotes node/graph/feature respectively. *MI-Estimator*: parameterized mutual information estimator. *Proj/Pred*: additional (MLP) projector or predictor. *Asymmetric*: asymmetric architectures such as EMA and Stop-Gradient, or two separate encoders for two branches. *Neg examples*: requiring negative examples to prevent trivial solutions. *Space* denotes space requirement for storing all the pairs. Our method is simple without any listed component and memory-efficient.

| | Methods | Target | MI-Estimator | Proj/Pred | Asymmetric | Neg examples | Space |
|---|---|---|---|---|---|---|---|
| Instance-level | DGI [48] | N-G | ✓ | - | - | ✓ | $O(N)$ |
| | MVGRL [15] | N-G | ✓ | - | ✓ | ✓ | $O(N)$ |
| | GRACE [57] | N-N | - | ✓ | - | ✓ | $O(N^2)$ |
| | GCA [58] | N-N | - | ✓ | - | ✓ | $O(N^2)$ |
| | BGRL [39] | N-N | - | ✓ | ✓ | - | $O(N)$ |
| | CCA-SSG (Ours) | F-F | - | - | - | - | $O(D^2)$ |

targeting a negative-sample-free approach for GNN learning through asymmetric architectures [12, 6]. However, it requires additional components, e.g., an exponential moving average (EMA) and Stop-Gradient, to empirically avoid degenerated solutions, leading to a more intricate architecture.

Deviating from the large body of previous works on contrastive learning, in this paper we take a new perspective to address SSL on graphs. We introduce Canonical Correlation Analysis inspired Self-Supervised Learning on Graphs (CCA-SSG), a simple yet effective approach that opens the way to a new SSL objective and frees the model from intricate designs. It follows the common practice of prior arts, generating two views of an input graph through random augmentation and acquiring node representations through a shared GNN encoder. Differently, we propose to harness a *non-contrastive* and *non-discriminative* feature-level objective, which is inspired by the well-studied Canonical Correlation Analysis (CCA) methods [18, 10, 11, 14, 2, 4]. More specifically, the new objective aims at maximizing the correlation between two augmented views of the same input and meanwhile decorrelating different (feature) dimensions of a single view's representation. We show that the objective 1) essentially pursuits discarding augmentation-variant information and preserving augmentation-invariant information, and 2) can prevent dimensional collapse [19] (i.e., different dimensions capture the same information) in nature. Furthermore, our theoretical analysis sheds more lights that under mild assumptions, our model is an instantiation of Information Bottleneck Principle [43, 44, 37] under SSL settings [53, 9, 45].

To sum up, as shown in Table 1, our new objective induces a simple and light model without reliance on negative pairs [48, 15, 57, 58], a parameterized mutual information estimator [48, 15], an additional projector or predictor [57, 58, 39] or asymmetric architectures [39, 15]. We provide a thorough evaluation for the model on seven node classification benchmarks. The empirical results demonstrate that despite its simplicity, CCA-SSG can achieve very competitive performance in general and even superior test accuracy in five datasets. It is worth noting that our approach is agnostic to the input data format, which means that it can potentially be applied to other scenarios beyond graph-structured data (such as vision, language, etc.). We leave such a technical extension for future works.

**Our contributions are as follows:**

**1)** We introduce a non-contrastive and non-discriminative objective for self-supervised learning, which is inspired by Canonical Correlation Analysis methods. It does not rely on negative samples, and can naturally remove the complicated components. Based on it we propose CCA-SSG, a simple yet effective framework for learning node representations without supervision (see Section 3).

**2)** We theoretically prove that the proposed objective aims at keeping augmentation-invariant information while discarding augmentation-variant one, and possesses an inherent relationship to an embodiment of Information Bottleneck Principle under self-supervised settings (see Section 4).

**3)** Experimental results show that without complex designs, our method outperforms state-of-the-art self-supervised methods MVGRL [15] and GCA [58] on 5 out of 7 benchmarks. We also provide thorough ablation studies on the effectiveness of the key components of CCA-SSG (see Section 5).

## 2   Related Works and Background

**Contrastive Learning on Graphs.**   Contrastive methods [46, 40, 17, 16, 5, 12] have been shown to be effective for unsupervised learning in vision, which have also been adapted to graphs. Inspired by the local-global mutual information maximization viewpoints [17], DGI [48] and InfoGraph [38] put forward unsupervised schemes for node and graph representation learning, respectively. MVGRL [15] generalizes CMC [40] to graph-structured data by introducing graph diffusion [23] to create another view for a graph. GCC [33] adopts InfoNCE loss [46] and MoCo-based negative pool [16] for large-scale GNN pretraining. GRACE [57], GCA [58] and GraphCL [52] follow the spirit of SimCLR [5] and learn node/graph representations by directly treating other nodes/graphs as negative samples. BGRL [39] targets a negative-sample-free model, inspired by BYOL [12], on node representation learning. But it still requires complex asymmetric architectures.

**Feature-level Self-supervised Objectives.**   The above-mentioned methods all focus on instance-level contrastive learning. To address their drawbacks, some recent works have been turning to feature-level objectives. For example, Contrastive Clustering [25] regards different feature dimensions as different clusters, thus combining the cluster-level discrimination with instance-level discrimination. W-MSE [8] performs a differentiable whitening operation on learned embeddings, which implicitly scatters data points in embedding space. Barlow Twins [53] borrows the idea of redundancy reduction and adopts a soft decorrelation term that makes the cross-correlation matrix of two views' representations close to an identity matrix. By contrast, our method is based on the classical Canonical Correlation Analysis, working by correlating the representations of two views from data augmentation and meanwhile decorrelating different feature dimensions of each view's representation.

**Canonical Correlation Analysis.**   CCA is a classical multivariate analysis method, which is first introduced in [18]. For two random variables $X \in \mathbb{R}^m$ and $Y \in \mathbb{R}^n$, their covariance matrix is $\Sigma_{XY} = Cov(X, Y)$. CCA aims at seeking two vectors $a \in \mathbb{R}^m$ and $b \in \mathbb{R}^n$ such that the correlation $\rho = \mathrm{corr}(a^\top X, b^\top Y) = \frac{a^\top \Sigma_{XY} b}{\sqrt{a^\top \Sigma_{XX} a}\sqrt{b^\top \Sigma_{YY} b}}$ is maximized. Formally, the objective is

$$\max_{a,b} a^\top \Sigma_{XY} b, \text{ s.t. } a^\top \Sigma_{XX} a = b^\top \Sigma_{YY} b = 1. \tag{1}$$

For multi-dimensional cases, CCA seeks two sets of vectors maximizing their correlation and subjected to the constraint that they are uncorrelated with each other [10]. Later studies apply CCA to multi-view learning with deep models [2, 11, 14], by replacing the linear transformation with neural networks. Concretely, assuming $X_1, X_2$ as two views of an input data, it optimizes

$$\max_{\theta_1,\theta_2} \mathrm{Tr}\left(P_{\theta_1}^\top(X_1) P_{\theta_2}(X_2)\right) \text{ s.t. } P_{\theta_1}^\top(X_1) P_{\theta_1}(X_1) = P_{\theta_2}^\top(X_2) P_{\theta_2}(X_2) = I. \tag{2}$$

where $P_{\theta_1}$ and $P_{\theta_2}$ are two feedforward neural networks and $I$ is an identity matrix. Despite its preciseness, such computation is really expensive [4]. Fortunately, soft CCA [4] removes the hard decorrelation constraint by adopting the following Lagrangian relaxation:

$$\min_{\theta_1,\theta_2} \mathcal{L}_{dist}\left(P_{\theta_1}(X_1), P_{\theta_2}(X_2)\right) + \lambda\left(\mathcal{L}_{SDL}(P_{\theta_1}(X_1)) + \mathcal{L}_{SDL}(P_{\theta_2}(X_2))\right), \tag{3}$$

where $\mathcal{L}_{dist}$ measures correlation between two views' representations and $\mathcal{L}_{SDL}$ (called stochastic decorrelation loss) computes an $L_1$ distance between $P_{\theta_i}(X_i)$ and an identity matrix, for $i = 1, 2$.

## 3   Approach

### 3.1   Model Framework

In this paper we focus on self-supervised node representation learning, where we consider a single graph $\mathbf{G} = (\mathbf{X}, \mathbf{A})$. $\mathbf{X} \in \mathbb{R}^{N \times F}$ and $\mathbf{A} \in \mathbb{R}^{N \times N}$ denote node features and adjacency matrix respectively. Here $N$ is the number of nodes within the graph and $F$ denotes feature dimension.

Our model simply consists of three parts: 1) a random graph augmentation generator $\mathcal{T}$. 2) a GNN-based graph encoder $f_\theta$ where $\theta$ denotes its parameters. 3) a novel feature-level objective function based on Canonical Correlation Analysis. Fig. 1 is an illustration of the proposed model.

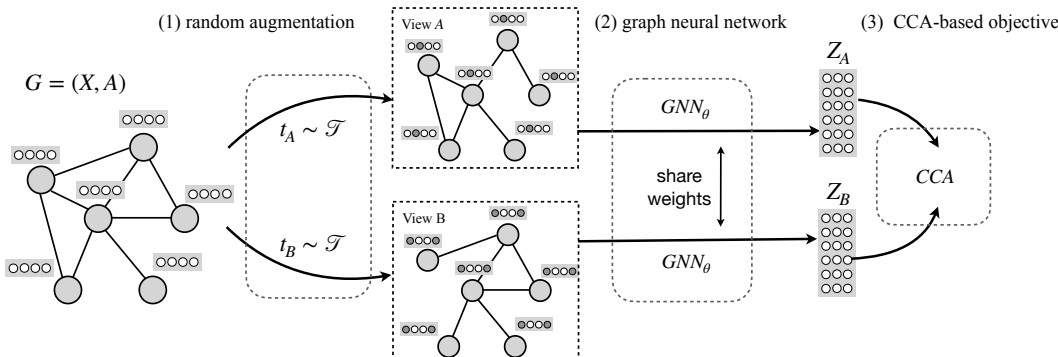

Figure 1: Illustration of the proposed model: given an input graph, we first generate two views through random augmentations: edge dropping and node feature masking. The two views are subsequently put into a shared GNN encoder to generate representations. The loss function is applied on the column-normalized embedding matrix of the two views. Note that this simple yet effective pipeline can also be conceptually applied for other data like vision and texts, which we leave for future works.

---

**Algorithm 1:** PyTorch-style code for CCA-SSG

```
# f: encoder network
# lambda: trade-off
# D: embedding dimension
# g: input graph
# feat: node features

# generate two views through random augmentation
g1, feat1 = augment(g, feat)
g2, feat2 = augment(g, feat)
z1 = f(g1, feat1) # embedding of the 1st view
z2 = f(g2, feat2) # embedding of the 2st view

# batch normalization
z1_norm = ((z1 - z1.mean(0)) / z1.std(0))/ sqrt(N)
z2_norm = ((z2 - z2.mean(0)) / z2.std(0))/ sqrt(N)

# covariance matrix of each view
c1 = torch.mm(z1_norm.T(), z1_norm)
c2 = torch.mm(z2_norm.T(), z2_norm)

iden = torch.eye(D)
loss_inv = (z1_norm - z2_norm).pow(2).sum()
loss_dec_1 = (c1 - iden).pow(2).sum()
loss_dec_2 = (c2 - iden).pow(2).sum()
loss_dec = loss_dec_1 + loss_dec_2
loss = loss_inv + lambda * loss_dec
```

**Graph augmentations**. We consider the standard pipeline for random graph augmentation that has been commonly used in previous works [57, 39]. To be specific, we harness two ways for augmentation: **edge dropping** and **node feature masking**. Edge dropping randomly drops a fraction of edges from the original graph, while node feature masking randomly masks a fraction of features for all the nodes. In this way, $\mathcal{T}$ is composed of all the possible graph transformation operations and each $t \sim \mathcal{T}$ denotes a specific graph transformation for graph $G$. Note that we use commonly adopted augmentation methods to stay our focus on the design of objective function and conduct fair comparison with existing approaches. More complicated random augmentations [52, 58] can also be readily plugged into our model. Details for the used augmentation functions are in Appendix E.

**Training**. In each training iteration, we first randomly sample two graph transformations $t_A$ and $t_B$ from $\mathcal{T}$, and then generate two views $\tilde{\mathbf{G}}_A = (\tilde{\mathbf{X}}_A, \tilde{\mathbf{A}}_A)$ and $\tilde{\mathbf{G}}_B = (\tilde{\mathbf{X}}_B, \tilde{\mathbf{A}}_B)$ according to the transformations. The two views are subsequently fed into a shared GNN encoder to generate the node embeddings of the two views: $\mathbf{Z}_A = f_\theta(\tilde{\mathbf{X}}_A, \tilde{\mathbf{A}}_A)$, $\mathbf{Z}_B = f_\theta(\tilde{\mathbf{X}}_B, \tilde{\mathbf{A}}_B)$, where $\mathbf{Z}_A, \mathbf{Z}_B \in \mathbb{R}^{N \times D}$ and $D$ denotes embedding dimension. We further normalize the node embeddings along instance dimension so that each feature dimension has a 0-mean and $1/\sqrt{N}$-standard deviation distribution:

$$\tilde{\mathbf{Z}} = \frac{\mathbf{Z} - \mu(\mathbf{Z})}{\sigma(\mathbf{Z}) * \sqrt{N}} \tag{4}$$

The normalized $\tilde{\mathbf{Z}}_A$, $\tilde{\mathbf{Z}}_B$ will be used to compute a feature-level objective in Section 3.2. To help better understand the proposed framework, we provide the PyTorch-style pseudocode for training CCA-SSG in Algorithm 1.

**Inference**. To generate node embeddings for downstream tasks, we put the original graph $\mathbf{G} = (\mathbf{X}, \mathbf{A})$ into the trained graph neural network $f_\theta$ and obtain node embeddings $\mathbf{Z} = f_\theta(\mathbf{X}, \mathbf{A})$.

## 3.2 Learning Objective

Canonical Correlation Analysis has shown its great power in multi-view learning like instance recognition [4]. However, it still remains unexplored to leverage CCA for self-supervised learning. Note that in SSL, one generates two sets of data from the same input through transformation or random data augmentation, which could be regraded as two views of the input data. This inspires us to introduce the following objective for self-supervised representation learning:

$$\mathcal{L} = \underbrace{\left\| \tilde{\mathbf{Z}}_A - \tilde{\mathbf{Z}}_B \right\|_F^2}_{\text{invariance term}} + \lambda \underbrace{\left( \left\| \tilde{\mathbf{Z}}_A^\top \tilde{\mathbf{Z}}_A - \mathbf{I} \right\|_F^2 + \left\| \tilde{\mathbf{Z}}_B^\top \tilde{\mathbf{Z}}_B - \mathbf{I} \right\|_F^2 \right)}_{\text{decorrelation term}} \tag{5}$$

where $\lambda$ is a non-negative hyperparameter trading off two terms. Note that minimizing the invariance term is essentially maximizing the correlation between two views as their representations are already normalized. In SSL, as the two augmented views come randomly from the same distribution, we can adopt one encoder $f_\theta$ that is shared across two branches and seek for a regularization that encourages different feature dimensions to capture distinct semantics via the decorrelation term.

We next provide a variance-covariance perspective to the new objective, following similar lines of reasoning in [41, 42]. Assume that input data come from a distribution $\boldsymbol{x} \sim p(\boldsymbol{x})$ and $\boldsymbol{s}$ is a view of $\boldsymbol{x}$ through random augmentation $\boldsymbol{s} \sim p_{aug}(\cdot|\boldsymbol{x})$. Denote $\boldsymbol{z_s}$ as the representation of $\boldsymbol{s}$, then minimizing the invariance term, by expectation, is to minimize the variance of the normalized representation $\tilde{\boldsymbol{z}}_{\boldsymbol{s}}$, conditioned on $\boldsymbol{x}$. Also, minimizing the decorrelation term is to push the off-diagonal elements of the covariance matrix (given by two $\tilde{\boldsymbol{z}}_{\boldsymbol{s}}$'s) close to 0. Formally, we have

$$\mathcal{L}_{inv} = \left\| \tilde{\mathbf{Z}}_A - \tilde{\mathbf{Z}}_B \right\|_F^2 = \sum_{i=1}^{N} \sum_{k=1}^{D} (\tilde{z}_{i,j}^A - \tilde{z}_{i,j}^B)^2 \cong \mathbb{E}_{\boldsymbol{x}} \left[ \sum_{k=1}^{D} \mathbb{V}_{\boldsymbol{s}|\boldsymbol{x}}[\tilde{\boldsymbol{z}}_{s,k}] \right] * 2N, \tag{6}$$

$$\mathcal{L}_{dec} = \left\| \tilde{\mathbf{Z}}_S^\top \tilde{\mathbf{Z}}_S - \mathbf{I} \right\|_F^2 = \|\text{Cov}_{\boldsymbol{s}}[\tilde{\boldsymbol{z}}] - I\|_F^2 \cong \sum_{i \neq j} \left( \rho_{i,j}^{\boldsymbol{z}_s} \right)^2, \text{ for } \tilde{\mathbf{Z}}_S \in \{\tilde{\mathbf{Z}}_A, \tilde{\mathbf{Z}}_B\}, \tag{7}$$

where $\rho$ is the Pearson correlation coefficient.

## 3.3 Advantages over Contrastive Methods

In this subsection we provide a systematic comparison with previous self-supervised methods for node representation learning, including DGI [48], MVGRL [15], GRACE [57], GCA [58] and BGRL [39], and highlight the merits of CCA-SSG. A quick overview is presented in Table 1.

**No reliance on negative samples**. Most of previous works highly rely on negative pairs to avoid collapse or interchangeable, trivial/degenerated solutions [48, 15, 57, 58]. E.g., DGI and MVGRL generate negative examples by corrupting the graph structure severely, and GRACE/GCA treats all the other nodes within a graph as negative examples. However, for self-supervised learning on graphs, it is non-trivial to construct informative negative examples since nodes are structurally connected, and selecting negative examples in an arbitrary manner may lead to large variance for stochastic gradients and slow training convergence [51]. The recently proposed BGRL model adopts asymmetric encoder architectures for SSL on graphs without the use of negative samples. However, though BGRL could avoid collapse empirically, it still remains as an open problem concerning its theoretical guarantee for preventing trivial solutions [41]. Compared with these methods, our model does not rely on negative pairs and asymmetric encoders. The feature decorrelation term can naturally prevent trivial solutions caused by the invariance term. We discuss the collapse issue detailedly in Appendix B.

**No MI estimator, projector network nor asymmetric architectures**. Most previous works rely on additional components besides the GNN encoder to estimate some score functions in final objectives. DGI and MVGRL require a parameterized estimator to approximate mutual information between two views, and GRACE leverages a MLP projector followed by an InfoNCE estimator. BGRL harnesses asymmetric encoder architecture which consists of EMA (Exponential Moving Average), Stop-Gradient and an additional projector. MVGRL also induces asymmetric architectures as it adopts two different GNNs for the input graph and the diffusion graph respectively. In contrast, our approach requires no additional components except a single GNN encoder.

**Better efficiency and scalability to large graphs**. Consider a graph with $N$ nodes. DGI and MVGRL contrast node embeddings with graph embedding, which would require $O(N)$ space cost.

GRACE treats two views of the same node as positive pairs and treat views of different nodes as negative pairs, which would take $O(N^2)$ space. BGRL focuses only on positive pairs, which will also take $O(N)$ space. By contrast, our method works on feature dimension. If we embed each node into a $D$-dimensional vector, the computation of the loss function would require $O(D^2)$ space. This indicates that the memory cost does not grow consistently as the size of graph increases. As a result, our method is promising for handling large-scale graphs without prohibitively large space costs.

## 4 Theoretical Insights with Connection to Information Theory

In this section we provide some analysis of the proposed objective function: 1) Interpretation of the loss function with entropy and mutual information. 2) The connection between the proposed objective and the Information Bottleneck principle. 3) Why the learned representations would be informative to downstream tasks. The proofs of propositions, theorems and corollaries are in Appendix D.

**Notations.** Denote the random variable of input data as $X$ and the downstream task as $T$ (it could be the label $Y$ if the downstream task is classification). Note that in SSL, we have no access to $T$ in training and here we introduce the notation for our analysis. Define $S$ as the self-supervised signal (i.e., an augmented view of $X$), and $S$ shares the same space as $X$. Our model learns a representation for the input, denoted by $Z_X$ and its views, denoted by $Z_S$. $Z_X = f_\theta(X), Z_S = f_\theta(S)$, $f_\theta(\cdot)$ is a encoder shared by the original data and its views, which is parameterized by $\theta$. The target of representation learning is to learn a optimal encoder parameter $\theta$. Furthermore, for random variable $A, B, C$, we use $I(A, B)$ to denote the mutual information between $A$ and $B$, $I(A, B|C)$ to denote conditional mutual information of $A$ and $B$ on a given $C$, $H(A)$ for the entropy, and $H(A|B)$ for conditional entropy. The proofs of propositions, theorems and corollaries are in Appendix D.

### 4.1 An Entropy and Mutual Information Interpretation of the Objective

We first introduce an assumption about the distributions of $P(Z_S)$ and $P(Z_S|X)$.

**Assumption 1.** *(Gaussian assumption of $P(Z_S|X)$ and $P(Z_S)$):*

$$P(Z_S|X) = \mathcal{N}(\mu_X, \Sigma_X), P(Z_S) = \mathcal{N}(\mu, \Sigma). \tag{8}$$

With Assumption 1, we can arrive at the following propositions:

**Proposition 1.** *In expectation, minimizing Eq. (6) is equivalent to minimizing the entropy of $Z_S$ conditioned on input $X$, i.e.,*

$$\min_\theta \mathcal{L}_{inv} \cong \min_\theta H(Z_S|X). \tag{9}$$

**Proposition 2.** *Minimizing Eq. (7) is equivalent to maximizing the entropy of $Z_S$, i.e.,*

$$\min_\theta \mathcal{L}_{dec} \cong \max_\theta H(Z_S). \tag{10}$$

The two propositions unveil the effects of two terms in our objective. Combining two propositions, we can further interpret Eq. (5) from an information-theoretic perspective.

**Theorem 1.** *By optimizing Eq (5), we maximize the mutual information between the augmented view's embedding $Z_S$ and the input data $X$, and minimize the mutual information between $Z_S$ and the view itself $S$, conditioned on the input data $X$. Formally we have*

$$\min_\theta \mathcal{L} \Rightarrow \max_\theta I(Z_S, X) \text{ and } \min_\theta I(Z_S, S|X). \tag{11}$$

The proof is based on the facts $I(Z_S, X) = H(Z_S) - H(Z_S|X)$ and $I(Z_S, S|X) = H(Z_S|X) + H(Z_S|S) = H(Z_S|X)$. Theorem 1 indicates that our objective Eq. (5) learns representations that maximize the information of the input data, i.e., $I(Z_S, X)$, and meanwhile minimize the lost information during augmentation, i.e., $I(Z_S, S|X)$.

### 4.2 Connection with the Information Bottleneck Principle

The analysis in Section 4.1 enables us to further build a connection between our objective Eq. (5) and the well-studied Information Bottleneck Principle [43, 44, 37, 1] under SSL settings. Recall that the supervised Information Bottleneck (IB) is defined as follows:

**Definition 1.** *The supervised IB aims at maximizing an Information Bottleneck Lagrangian:*

$$\mathcal{IB}_{sup} = I(Y, Z_X) - \beta I(X, Z_X), \text{ where } \beta > 0. \tag{12}$$

As we can see, $\mathcal{IB}_{sup}$ attempts to maximize the information between the data representation $Z_X$ and its corresponding label $Y$, and concurrently minimize the information between $Z_X$ and the input data $X$ (i.e., exploiting compression of $Z_X$ from $X$). The intuition of IB principle is that $Z_X$ is expected to contain only the information that is useful for predicting $Y$.

Several recent works [9, 45, 53] propose various forms of IB under self-supervised settings. The most relevant one names Self-supervised Information Bottleneck:

**Definition 2.** *(Self-supervised Information Bottleneck [53]). The Self-supervised IB aims at maximizing the following Lagrangian:*

$$\mathcal{IB}_{ssl} = I(X, Z_S) - \beta I(S, Z_S), \text{ where } \beta > 0. \tag{13}$$

Intuitively, $\mathcal{IB}_{ssl}$ posits that a desirable representation is expected to be informative to augmentation invariant features, and to be a maximally compressed representation of the input.

Our objective Eq. (5) is essentially an embodiment of $\mathcal{IB}_{ssl}$:

**Theorem 2.** *Assume $0 < \beta \leq 1$, then by minimizing Eq. (5), the self-supervised Information Bottleneck objective is maximized, formally:*

$$\min_{\theta} \mathcal{L} \Rightarrow \max_{\theta} \mathcal{IB}_{ssl} \tag{14}$$

Theorem 2 also shows that Eq. (5) implicitly follows the same spirit of IB principle under self-supervised settings. As further enlightenment, we can relate Eq. (5) with the *multi-view Information Bottleneck* [9] and the *minimal and sufficient representations for self-supervision* [45]:

**Corollary 1.** *Let $X_1 = S$, $X_2 = X$ and assume $0 < \beta \leq 1$, then minimizing Eq. (5) is equivalent to minimizing the Multi-view Information Bottleneck Loss in [9]:*

$$\mathcal{L}_{MIB} = I(Z_1, X_1 | X_2) - \beta I(X_2, Z_1), \text{ where } 0 < \beta \leq 1. \tag{15}$$

**Corollary 2.** *When the data augmentation process is reversible, minimizing Eq. (5) is equivalent to learning the Minimal and Sufficient Representations for Self-supervision in [45]:*

$$Z_X^{ssl} = \arg\max_{Z_X} I(Z_X, S), Z_X^{ssl_{min}} = \arg\min_{Z_X} H(Z_X | S) \text{ s.t. } I(Z_X, S) \text{ is maximized.} \tag{16}$$

### 4.3 Influence on Downstream Tasks

We have provided a principled understanding for our new objective. Next, we discuss its effect on downstream tasks $T$. The rationality of data augmentations in SSL is rooted in a conjecture that an ideal data augmentation approach would not change the information related to its label. We formulate this hypothesis as a building block for analysis on downstream tasks [36, 9].

**Assumption 2.** *(Task-relevant information and data augmentation). All the task-relevant information is shared across the input data $X$ and its augmentations $S$, i.e., $I(X, T) = I(S, T) = I(X, S, T)$, or equivalently, $I(X, T | S) = I(S, T | X) = 0$.*

This indicates that all the task-relevant information is contained in augmentation invariant features. We proceed to derive the following theorem which reveals the efficacy of the learned representations by our objective with respect to downstream tasks.

**Theorem 3.** *(Task-relevant/irrelevant information). By optimizing Eq. (5), the task-relevant information $I(Z_S, T)$ is maximized, and the task-irrelevant information $H(Z_S | T)$ is minimized. Formally,*

$$\min_{\theta} \mathcal{L} \Rightarrow \max_{\theta} I(Z_S, T) \text{ and } \min_{\theta} H(Z_S | T). \tag{17}$$

Therefore, the learned representation $Z_S$ is expected to contain minimal and sufficient information about downstream tasks [45, 9], which further illuminates the reason why the embeddings given by SSL approaches have superior performance on various downstream tasks.

Table 2: Test accuracy on citation networks. The *input* column highlights the data used for training. ($\mathbf{X}$ for node features, $\mathbf{A}$ for adjacency matrix, $\mathbf{S}$ for diffusion matrix, and $\mathbf{Y}$ for node labels).

|  | Methods | Input | Cora | Citeseer | Pubmed |
|---|---|---|---|---|---|
| Supervised | MLP [47] | $\mathbf{X}, \mathbf{Y}$ | 55.1 | 46.5 | 71.4 |
|  | LP [56] | $\mathbf{A}, \mathbf{Y}$ | 68.0 | 45.3 | 63.0 |
|  | GCN [22] | $\mathbf{X}, \mathbf{A}, \mathbf{Y}$ | 81.5 | 70.3 | 79.0 |
|  | GAT [47] | $\mathbf{X}, \mathbf{A}, \mathbf{Y}$ | $83.0 \pm 0.7$ | $72.5 \pm 0.7$ | $79.0 \pm 0.3$ |
| Unsupervised | Raw Features [48] | $\mathbf{X}$ | $47.9 \pm 0.4$ | $49.3 \pm 0.2$ | $69.1 \pm 0.3$ |
|  | Linear CCA [18] | $\mathbf{X}$ | $58.9 \pm 1.5$ | $27.5 \pm 1.3$ | $75.8 \pm 0.4$ |
|  | DeepWalk [32] | $\mathbf{A}$ | $70.7 \pm 0.6$ | $51.4 \pm 0.5$ | $74.3 \pm 0.9$ |
|  | GAE [21] | $\mathbf{X}, \mathbf{A}$ | $71.5 \pm 0.4$ | $65.8 \pm 0.4$ | $72.1 \pm 0.5$ |
|  | DGI [48] | $\mathbf{X}, \mathbf{A}$ | $82.3 \pm 0.6$ | $71.8 \pm 0.7$ | $76.8 \pm 0.6$ |
|  | MVGRL[1] [15] | $\mathbf{X}, \mathbf{S}, \mathbf{A}$ | $83.5 \pm 0.4$ | $\mathbf{73.3 \pm 0.5}$ | $80.1 \pm 0.7$ |
|  | GRACE[2] [57] | $\mathbf{X}, \mathbf{A}$ | $81.9 \pm 0.4$ | $71.2 \pm 0.5$ | $80.6 \pm 0.4$ |
|  | CCA-SSG (Ours) | $\mathbf{X}, \mathbf{A}$ | $\mathbf{84.2 \pm 0.4}$ | $73.1 \pm 0.3$ | $\mathbf{81.6 \pm 0.4}$ |

[1] Results on Cora with authors' code is inconsistent with [15]. We adopt the results with authors' code.
[2] Results are from our reproducing with authors' code, as [57] did not use the public splits.

## 5 Experiments

We assess the quality of representations after self-supervised pretraining on seven node classification benchmarks: *Cora, Citeseer, Pubmed, Coauthor CS, Coauthor Physics* and *Amazon Computer, Amazon-Photo*. We adopt the public splits for *Cora, Citeseer, Pubmed*, and a 1:1:9 training/validation/testing splits for the other 4 datasets. Details of the datasets are in Appendix E.

**Evaluation protocol**. We follow the linear evaluation scheme as introduced in [48]: **i)** We first train the model on all the nodes in a graph without supervision, by optimizing the objective in Eq. (5). **ii)** After that, we freeze the parameters of the encoder and obtain all the nodes' embeddings, which are subsequently fed into a linear classifier (i.e., a logistic regression model) to generate a predicted label for each node. In the second stage, only nodes in training set are used for training the classifier, and we report the classification accuracy on testing nodes.

We implement the model with PyTorch. All experiments are conducted on a NVIDIA V100 GPU with 16 GB memory. We use the Adam optimizer [20] for both stages. The graph encoder $f_\theta$ is specified as a standard two-layer GCN model [22] for all the datasets except *citeseer* (where we empirically find that a one-layer GCN is better). We report the mean accuracy with a standard deviation through 20 random initialization (on *Coauthor CS, Coauthor Physics* and *Amazon Computer, Amazon-Photo*, the split is also randomly generated). Detailed hyperparameter settings are in Appendix E.

### 5.1 Comparison with Peer Methods

We compare CCA-SSG with classical unsupervised models, Deepwalk [32] and GAE [21], and self-supervised models, DGI [48], MVGRL [15], GRACE [57] and GCA [58]. We also compare with supervised learning models, including MLP, Label Propagation (LP) [56], and supervised baselines GCN [22] and GAT [47][3]. The results of baselines are quoted from [15, 57, 58] if not specified.

We report the node classification results of citation networks and other datasets in Table 2 and Table 3 respectively. As we can see, CCA-SSG outperforms both the unsupervised competitors and the fully supervised baselines on *Cora* and *Pubmed*, despite its simple architecture. On *Citeseer*, CCA-SSG achieves competitive results as of the most powerful baseline MVGRL. On four larger benchmarks, CCA-SSG also achieves the best performance in four datasets except *Coauther-Physics*. It is worth mentioning that we empirically find that on *Coauthor-CS* a pure 2-layer-MLP encoder is better than GNN models. This might because the graph-structured information is much less informative than the node features, presumably providing harmful signals for classification (in fact, on *Coauthor-CS*, linear models using merely node features can greatly outperform DeepWalk/DeepWalk+features).

---

[3]The BGRL [39] is not compared as its source code has not been released.

Table 3: Test accuracy on co-author and co-purchase networks. We report both mean accuracy and standard deviation. Results of baseline models are from [58].

| | Methods | Input | Computer | Photo | CS | Physics |
|---|---|---|---|---|---|---|
| | Supervised GCN [22] | $\mathbf{X}, \mathbf{A}, \mathbf{Y}$ | $86.51 \pm 0.54$ | $92.42 \pm 0.22$ | $93.03 \pm 0.31$ | $95.65 \pm 0.16$ |
| | Supervised GAT [47] | $\mathbf{X}, \mathbf{A}, \mathbf{Y}$ | $86.93 \pm 0.29$ | $92.56 \pm 0.35$ | $92.31 \pm 0.24$ | $95.47 \pm 0.15$ |
| Unsupervised | Raw Features [48] | $\mathbf{X}$ | $73.81 \pm 0.00$ | $78.53 \pm 0.00$ | $90.37 \pm 0.00$ | $93.58 \pm 0.00$ |
| | Linear CCA [18] | $\mathbf{X}$ | $79.84 \pm 0.53$ | $86.92 \pm 0.72$ | $93.13 \pm 0.18$ | $95.04 \pm 0.17$ |
| | DeepWalk [32] | $\mathbf{A}$ | $85.68 \pm 0.06$ | $89.44 \pm 0.11$ | $84.61 \pm 0.22$ | $91.77 \pm 0.15$ |
| | DeepWalk + features | $\mathbf{X}, \mathbf{A}$ | $86.28 \pm 0.07$ | $90.05 \pm 0.08$ | $87.70 \pm 0.04$ | $94.90 \pm 0.09$ |
| | GAE [21] | $\mathbf{X}, \mathbf{A}$ | $85.27 \pm 0.19$ | $91.62 \pm 0.13$ | $90.01 \pm 0.71$ | $94.92 \pm 0.07$ |
| | DGI [48] | $\mathbf{X}, \mathbf{A}$ | $83.95 \pm 0.47$ | $91.61 \pm 0.22$ | $92.15 \pm 0.63$ | $94.51 \pm 0.52$ |
| | MVGRL [15] | $\mathbf{X}, \mathbf{S}, \mathbf{A}$ | $87.52 \pm 0.11$ | $91.74 \pm 0.07$ | $92.11 \pm 0.12$ | $95.33 \pm 0.03$ |
| | GRACE[1] [57] | $\mathbf{X}, \mathbf{A}$ | $86.25 \pm 0.25$ | $92.15 \pm 0.24$ | $92.93 \pm 0.01$ | $95.26 \pm 0.02$ |
| | GCA[1] [58] | $\mathbf{X}, \mathbf{A}$ | $87.85 \pm 0.31$ | $92.49 \pm 0.09$ | $93.10 \pm 0.01$ | $\mathbf{95.68 \pm 0.05}$ |
| | CCA-SSG (Ours) | $\mathbf{X}, \mathbf{A}$ | $\mathbf{88.74 \pm 0.28}$ | $\mathbf{93.14 \pm 0.14}$ | $\mathbf{93.31 \pm 0.22}$ | $95.38 \pm 0.06$ |

[1] GCA is essentially an enhanced version of GRACE by adopting adaptive augmentations. Both GRACE and GCA would suffer from *out of memory* on *Coauthor-Physics* using a GPU wth 16GB memory. The reported results are from authors' papers using a 32GB GPU.

## 5.2 Ablation Study and Scalability Comparison

**Effectiveness of invariance/decorrelation terms**. We alter our loss by removing the invariance/decorrelation term respectively to study the effects of each component, with results reported in Table 4. We find that only using the invariance term will lead to merely performance drop instead of completely collapsed solutions. This is because node embeddings are normalized along the instance dimension to have a zero-mean and fixed-standard deviation, and the worst solution is no worse than dimensional collapse (i.e., all the embeddings lie in an line, and our decorrelation term can help to prevent it) instead of complete collapse (i.e., all the embeddings degenerate into a single point). As expected, only optimizing the decorrelation term will lead to poor result, as the model learns nothing meaningful but disentangled representation. In Appendix B we discuss the relationship between complete/dimensional collapse, when the two cases happen and how to avoid them.

**Effect of decorrelation intensity**. We study how the intensity of feature decorrelation improves/degrades the performance by increasing the trade-off hyper-parameter $\lambda$. Fig. 2 shows test accuracy w.r.t. different $\lambda$'s on *Cora, Citeseer* and *Pubmed*. The performance benefits from a proper selection of $\lambda$ (from 0.0005 to 0.001 in our experiments). When $\lambda$ is too small, the decorrelation term does not work; if it is too large, the invariance term would be neglected, leading to serious performance degrade. An interesting finding is that even when $\lambda$ is very small or even equals to 0 (w/o $\mathcal{L}_{dec}$ in Table 4), the test accuracy on *Citeseer* does not degrade as much as that on *Cora* and *Citeseer*. The reason is that node embeddings of *Citeseer* is already highly uncorrelated even without the decorrelation term. Appendix F visualizes the correlation matrices without/with decorrelations.

**Effect of embedding dimension**. Fig. 3 shows the effect of the embedding dimension. Similar to contrastive methods [48, 15, 57, 58], CCA-SSG benefits from a large embedding dimension (compared with supervised learning), while the optimal embedding dimension of CCA-SSG (512 on most benchmarks) is a bit larger than other methods (usually 128 or 256). Yet, we notice a performance drop as the embedding dimension increases. We conjecture that the CCA is essentially a dimension-reduction method, the ideal embedding dimension ought to be smaller than the dimension of input. Hence we do not apply it on well-compressed datasets (e.g. ogbn-arXiv and ogbn-product).

**Scalability Comparison.** Table 5 compares model size, training time (till the epoch that gives the highest evaluation accuracy) and memory cost of CCA-SSG with other methods, on *Cora, Pubmed* and *Amazon-Computers*. Overall, our method has fewer parameters, shorter training time, and fewer memory cost than MVGRL, GRACE and GCA in most cases. DGI is another simple and efficient model, but it yields much poorer performance. The results show that despite its simplicity and efficiency, our method achieves even better (or competitive) performance.

Table 4: Ablation study of node classification accuracy (%) on the key components of CCA-SSG.

| Variants | Cora | Citeseer | Pubmed |
|---|---|---|---|
| Baseline | 84.2 | 73.1 | 81.6 |
| w/o $\mathcal{L}_{dec}$ | 79.1 | 72.2 | 75.3 |
| w/o $\mathcal{L}_{inv}$ | 40.1 | 28.9 | 46.5 |

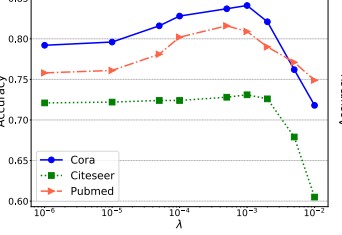 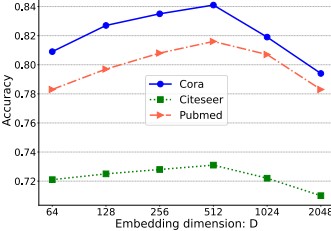

Figure 2: Effect of $\lambda$.  Figure 3: Effect of $D$.

Table 5: Comparison of the number of parameters, training time for achieving the best performance, and the memory cost of different methods on *Cora, Pubmed* and *Amazon-Computer*. MVGRL on Pubmed and Computer requires subgraph sampling with graph size $4000$. Others are full-graph.

| Methods | Cora ($N$: 2,708) | | | Pubmed ($N$: 19,717) | | | Computer ($N$: 13,752) | | |
|---|---|---|---|---|---|---|---|---|---|
| | #Paras | Time | Mem | #Paras | Time | Mem | #Paras | Time | Mem |
| DGI | 1260K | 6.4s | 1.4G | 782K | 5.9s | 1.9G | 919K | 14.1s | 1.9G |
| MVGRL | 1731K | 26.9s | 4.6G | 775K | 29s | 5.4G | 1049K | 31.5s | 5.5G |
| GRACE/GCA | 997K | 8.3s | 1.7G | 520K | 756s | 12.6G | 273K | 314s | 7.6G |
| CCA-SSG(Ours) | 997K | 3.8s | 1.6G | 519K | 9.6s | 2.7G | 656K | 14.8s | 2.5G |

## 6 Conclusion and Discussions

In this paper, we have introduced CCA-SSG, a conceptually simple, efficient yet effective method for self-supervised representation learning on graphs, based on the idea of Canonical Correlation Analysis. Compared with contrastive methods, our model does not require additional components except random augmentations and a GNN encoder, whose effectiveness is justified in experiments.

**Limitations of the work.** Despite the theoretical grounds and the promising experimental justifications, our method would suffer from several limitations. 1) The objective Eq. (5) is essentially performing dimension reduction, while SSL approach usually requires a large embedding dimension. As a result, our method might not work well on datasets where input data does not have a large feature dimension. 2) Like other augmentation based methods, CCA-SSG highly relies on a high-quality, informative and especially, label-invariant augmentations. However, the augmentations used in our model might not perfectly meet these requirements, and it remains an open problem how to generate informative graph augmentations that have non-negative impacts on the downstream tasks.

**Potential negative societal impacts.** This work explores a simple pipeline for representation learning without large amount of labeled data. However, in industry there are many career workers whose responsibility is to label or annotate data. The proposed method might reduce the need for labeling data manually, and thus makes a few individuals unemployed (especially for developing countries and remote areas). Furthermore, our model might be biased, as it tends to pay more attention to the majority and dominant features (shared information across most of the data). The minority group whose features are scare are likely to be downplayed by the algorithm.

## Acknowledgments and Disclosure of Funding

This work was supported in part by NSF under grants III-1763325, III-1909323, III-2106758, and SaTC-1930941. Qitian Wu and Junchi Yan were partly supported by Shanghai Municipal Science and Technology Major Project (2021SHZDZX0102). We thank Amazon Web Services for sponsoring computation resources for this work.

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
