# A  Algorithm

We provide the pseudo code for our method in Algorithm 2, the detailed description of which is in Section 3.1.

---

**Algorithm 2:** Algorithm for CCA-SSG

---

**Input:** A graph $\mathbf{G} = (\mathbf{X}, \mathbf{A})$ with $N$ nodes, where $\mathbf{X}$ is node feature matrix, and $\mathbf{A}$ is the adjacency matrix. Augmentations $\mathcal{T}$, encoder $f_\theta$ by random initialization, trade-off hyper-parameter $\lambda$, maximum training steps $T$.
**Output:** Learned encoder $f_\theta$.

---

1 **while** *not reaching* $T$ **do**
2     Sample two augmentation functions $t_A, t_B \sim \mathcal{T}$ ;
3     Generate transformed graphs: $\tilde{\mathbf{G}}_A = (\tilde{\mathbf{X}}_A, \tilde{\mathbf{A}}_A), \tilde{\mathbf{G}}_B = (\tilde{\mathbf{X}}_B, \tilde{\mathbf{A}}_B)$ ;
4     Get node embeddings through the graph neural network as encoder:
      $\mathbf{Z}_A = f_\theta(\tilde{\mathbf{X}}_A, \tilde{\mathbf{A}}_A), \mathbf{Z}_B = f_\theta(\tilde{\mathbf{X}}_B, \tilde{\mathbf{A}}_B)$ ;
5     Normalize embeddings along instance dimension: $\tilde{\mathbf{Z}}_A = \frac{\mathbf{Z}_A - \mu(\mathbf{Z}_A)}{\sigma(\mathbf{Z}_A) * \sqrt{N}}, \tilde{\mathbf{Z}}_B = \frac{\mathbf{Z}_B - \mu(\mathbf{Z}_B)}{\sigma(\mathbf{Z}_B) * \sqrt{N}}$ ;
6     Calculate the loss function $\mathcal{L}$ according to Eq. (5) ;
7     Update $\theta$ by gradient descent;
8 **Inference**: $\mathbf{Z} = f_\theta(\mathbf{X}, \mathbf{A})$, where $\theta$ is the frozen parameters of the encoder.

---

# B  Discussions on Degenerated Solutions in SSL

In this section we provide an illustration and some discussions for degenerated (collapsed) solutions, or namely trivial solutions, in self-supervised representation learning. The discussion is inspired by the separation of complete collapse and dimensional collapse proposed in [19]. We show that our method naturally avoids complete collapse through feature-wise normalization, and could prevent/alleviate dimensional collapse through the decorrelation term Eq. (7).

In most contrastive learning methods especially the augmentation-based ones [46, 16, 5, 40], both positive pairs and negative pairs are required for learning a model. For instance, the widely adopted InfoNCE [46] loss has the following formulation:

$$\mathcal{L}_{\text{InfoNCE}} = -\log \frac{\exp\left(z_i^A \cdot z_i^B / \tau\right)}{\sum_j \exp\left(z_i^A \cdot z_j^B / \tau\right)}, \tag{18}$$

where $z_i^A$ and $z_i^B$ are the (normalized) embeddings of two views of the same instance $i$, and $\tau$ is the temperature hyperparameter. The numerator enforces similarity between positive pairs (two views of the same instance), while the denominator promotes dis-similarity between negative pairs (two views of different instances). Therefore, minimizing Eq. (18) is equivalent to maximizing the cosine similarity of positive pairs and meanwhile minimizing the cosine similarity of negative pairs. Note that the normalization is applied for each instance (projecting the embedding onto a hypersphere), so we are essentially minimizing the distances between positive pairs and maximizing the distance between negative pairs. The previous work [50] provides a thorough analysis on the behaviors of the objective by decomposing it into two terms: 1) alignment term (for positive pairs) and 2) uniformity term (for negative pairs).

The alignment loss is defined as the expected distance between positive pairs:

$$\mathcal{L}_{\text{align}} \triangleq \mathbb{E}_{(x,y) \sim p_{\text{pos}}} \|f(x) - f(y)\|_2^\alpha, \text{ with } \alpha > 0. \tag{19}$$

The uniformity loss is the logarithm of the average pairwise Gaussian potential:

$$\mathcal{L}_{\text{uniformity}} \triangleq \log \mathbb{E}_{x,y \sim p_{\text{data}}} e^{-t\|f(x) - f(y)\|_2^2}, \text{ with } t > 0 \tag{20}$$

Intuitively, the alignment term makes the positive pairs close to each other on the hypersphere, while the uniformity term makes different data points distribute on the hypersphere uniformly.

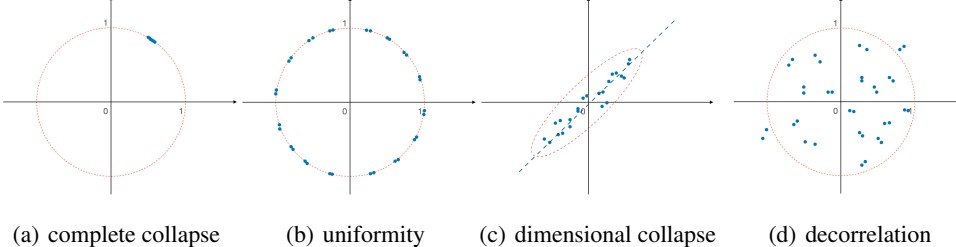

(a) complete collapse      (b) uniformity      (c) dimensional collapse      (d) decorrelation

Figure 4: An illustration of the two types of collapse and how to deal with them with a 2-d case. Blue circles are data points. Fig. 4(a): complete collapse, when all data samples degenerate to a same point on the hypersphere. Fig. 4(b): the uniformity loss keeps positive pairs close, but forces all data points to distribute on the hypersphere uniformly. Fig. 4(c): in dimensional collapse, the data points are not projected onto the hypersphere, but they distribute nearly as a line in the space, making them hard to discriminate. Fig. 4(d): the decorrelation term prevents dimensional collapse by directly decorrelating each dimensional representations, which implicitly scatters the data points.

In particular, only considering the alignment term in Eq. (19) will lead to trivial solutions: all the embeddings would degenerate to a fixed point on the hypersphere. This phenomenon is called **complete collapse** [19]. Denote $\mathbf{Z}_A$ and $\mathbf{Z}_B$ as two embedding matrix of two views ($\mathbf{Z} \in \mathbb{R}^{N \times D}$ and is row normalized), then in this case $\mathbf{Z}_A \mathbf{Z}_B^\top \cong \mathbf{1}$ is an all-one matrix (so as $\mathbf{Z}_A \mathbf{Z}_A^\top$ and $\mathbf{Z}_B \mathbf{Z}_B^\top$).

The uniformity term in Eq. (20) prevents complete collapse by separating the embeddings of arbitrary two data points, so that the data points would be embedded uniformly on the hypersphere. Fig. 4(a) and 4(b) provide an illustration for complete collapse and how the uniformity term prevents it.

Another kind of collapse that has been neglected by most existing works is **dimensional collapse** [19]. Different from complete collapse where all the data points degenerate into **a single point**, dimensional collapse means data points are distributed on **a line**, and each dimension captures exactly the same features (or different dimensions are highly correlated can capture the same information). Note that if the data representations are normalized along feature dimensions, all the data points would be projected onto a hypersphere. Under this circumstance there will not be dimensional collapse. However, if we normalize the output along the instance dimension so that each column has zero-mean and $1/\sqrt{N}$-standard deviation, as is done in this paper in Eq. (6), merely optimizing Eq. (6) would not prevent dimensional collapse, i.e. $\tilde{\mathbf{Z}}^\top \tilde{\mathbf{Z}} \cong \mathbf{1}$ ($\tilde{\mathbf{Z}} \in \mathbb{R}^{N \times D}$ is normalized by column).

In our model, the feature decorrelation term in Eq. (7) exactly prevents dimensional collapse by minimizing $\left\| \tilde{\mathbf{Z}}^\top \tilde{\mathbf{Z}} - \mathbf{I} \right\|_F^2$. Note that the diagonal term is always equal to 1, so we are pushing each dimension to capture orthogonal features. Also, the feature decorrelation term implicitly scatters the data points in the space, making them distinguishable for downstream tasks [19, 8]. An illustration of the dimensional collapse and the effect of feature decorrelation is provided in Fig. 4(c) and 4(d), respectively.

## C  Properties of Mutual Information and Entropy

In this section, we enumerate some useful properties of mutual information and entropy that will be used in Appendix D for proving the theorems. For any random variables $A, B, C, X, Z$, we have:

- **Property 1**. Non-negativity:

$$I(A, B) \geq 0, I(A, B|C) \geq 0. \tag{21}$$

- **Property 2**. Chain rule:

$$I(A, B, C) = I(A, B) - I(A, B|C). \tag{22}$$

- **Property 3**. Data Processing Inequality (DPI). $Z = f_\theta(X)$, then:

$$I(Z, A) = I(f_\theta(X), A) \leq I(X, A) \tag{23}$$

- **Property 4**. Non-negativity of discrete entropy. For discrete random variable:
$$H(A) \geq 0, H(A|B) \geq 0. \tag{24}$$
- **Property 5**. Relationship between entropy and mutual information:
$$H(A) = H(A|B) + I(A, B). \tag{25}$$
- **Property 6**. Entropy of deterministic function. If $Z$ is deterministic given $X$:
$$H(Z|X) = 0 \tag{26}$$
- **Property 7**. Entropy of Gaussian distribution. Assume $X$ obeys a $k$-dimensional Gaussian distribution, $X \sim \mathcal{N}(\mu, \Sigma)$, and we have
$$H(X) = \frac{k}{2}(\ln 2\pi + 1) + \frac{1}{2}\ln|\Sigma|. \tag{27}$$

# D   Proofs in Section 4

As already introduced in Section 4, we use $X$ and $S$ to denote the data and its augmentations respectively. We use $Z_X$ and $Z_S$ to denote their embeddings through the encoder $f_\theta$: $Z_X = f_\theta(X)$, $Z_S = f_\theta(S)$. We aim to learn the optimal encoder parameters $\theta$.

## D.1   Proof of Proposition 1

Restate Proposition 1:

**Proposition 1.** *In expectation, minimizing Eq. (6) is equivalent to minimizing the entropy of $Z_S$ conditioned on the input $X$, i.e.,:*
$$\min_\theta \mathcal{L}_{inv} \cong \min_\theta H(Z_S|X) \tag{28}$$

*Proof.* Assume input data come from a distribution $\boldsymbol{x} \sim p(\boldsymbol{x})$ and $\boldsymbol{s}$ is a view of $\boldsymbol{x}$ through random augmentation $\boldsymbol{s} \sim p_{aug}(\cdot|\boldsymbol{x})$. Denote $\boldsymbol{z_s}$ as the representation of $\boldsymbol{s}$. Note that $\boldsymbol{s}_1$ and and $\boldsymbol{s}_2$ both come from $p_{aug}(\cdot|\boldsymbol{x})$.

Recall the invariance term: $\mathcal{L}_{inv} = \left\|\tilde{\mathbf{Z}}_A - \tilde{\mathbf{Z}}_B\right\|_F^2 = \sum_{i=1}^N \sum_{k=1}^D \left(\tilde{z}_{i,k}^A - \tilde{z}_{i,k}^B\right)^2$. If we ignore the normalization and use $\boldsymbol{s}_1$ and $\boldsymbol{s}_2$ to represent view $A$ and view $B$. We have:

$$
\begin{aligned}
\mathcal{L}_{inv}/N \cong{}& \mathbb{E}_{\boldsymbol{x}}\left(\sum_{k=1}^D \mathbb{E}_{\boldsymbol{s}_1, \boldsymbol{s}_2 \sim p(\cdot|\boldsymbol{x})}(z_k^{\boldsymbol{s}_1} - z_k^{\boldsymbol{s}_2})^2\right) \\
={}& \mathbb{E}_{\boldsymbol{x}}\left(\sum_{k=1}^D \mathbb{E}_{\boldsymbol{s}_1, \boldsymbol{s}_2 \sim p(\cdot|\boldsymbol{x})}(z_k^{\boldsymbol{s}_1\,2} + z_k^{\boldsymbol{s}_2\,2} - 2*z_k^{\boldsymbol{s}_1} z_k^{\boldsymbol{s}_2})\right) \\
={}& 2*\mathbb{E}_{\boldsymbol{x}}\left(\sum_{k=1}^D \mathbb{V}_{\boldsymbol{s} \sim p(\cdot|\boldsymbol{x})} z_k^{\boldsymbol{s}}\right) \\
={}& 2*\sum_{k=1}^D \mathbb{E}_{\boldsymbol{x}}\left(\mathbb{V}_{\boldsymbol{s} \sim p(\cdot|\boldsymbol{x})} z_k^{\boldsymbol{s}}\right)
\end{aligned}
\tag{29}
$$

This indicates that minimizing $\mathcal{L}_{inv}$ is to minimize the variance of augmentation's representations conditioned on the input data.

Note the decorrelation term Eq. (7) aims to learn orthogonal representations at each dimension. If the representations are perfectly decorrelated, then $H(Z_S|X) = \sum_k H(Z_{S,k}|X)$. With Assumption 1, each dimensional representation also obeys a 1-dimensional Gaussian distribution, whose entropy is $H(Z_{S,k}|X) = \frac{1}{2}\log 2\pi e \sigma_k^{s\,2}$. This indicates by minimizing the variance of features at each dimension, its entropy is also minimized. Hence we have Proposition 1. $\qquad\square$

**Remark 1.** $I(Z_S, S|X) = H(Z_S|X) - H(Z_S|S, X) = H(Z_S|X)$ *(Property 6 in Appendix C). So $I(Z_S, S|X)$ is also minimized.*

### D.2 Proof of Proposition 2

Restate Proposition 2:

**Proposition 2.** *Minimizing Eq. (7) is equivalent to maximizing the entropy of $Z_S$, i.e.,*

$$\min_\theta \mathcal{L}_{dec} \cong \max_\theta H(Z_S). \tag{30}$$

*Proof.* With the assumption that $Z_S$ obeys a Gaussian distribution, we have:

$$\max_\theta H(Z_S) \cong \max_\theta \log |\Sigma_{Z_S}|, \tag{31}$$

where $|\Sigma_{Z_S}|$ is the determinant of the covariance matrix of the embeddings of the augmented data. Note that in our implementation we normalize the embedding matrix along the instance dimension: $\Sigma_{Z_S} \cong \tilde{\mathbf{Z}}_S^\top \tilde{\mathbf{Z}}_S$, so the diagonal entries of $\Sigma_{Z_S}$ are all 1's. And $\Sigma_{Z_S} \in \mathbb{R}^{D \times D}$ is a symmetric matrix.

If $\lambda_1, \lambda_2, \cdots, \lambda_D$ are the $D$ eigenvalues of $\Sigma_{Z_S}$, then $\sum_{i=1}^{D} \lambda_i = \text{trace}(\Sigma_{Z_S}) = D$. We have

$$\log |\Sigma_{Z_{\theta,X'}}| = \log \prod_{i=1}^{D} \lambda_i = \underbrace{\sum_{i=1}^{D} \log \lambda_i \leq D \log \frac{\sum_{i=1}^{D} \lambda_i}{D}}_{\text{Jensen Inequality}} = 0. \tag{32}$$

This means that the upper bound of $|\Sigma_{Z_S}|$ is 1, and the upper bound is achieved if and only if $\lambda_i = 1$ for $\forall i$, which indicates $\Sigma_{Z_S}$ is an identity matrix. This global optimum is exactly the same as that of the feature decorrelation term $\mathcal{L}_{dec}$ in Eq. (7). Therefore we conclude the proof. $\qquad \square$

### D.3 Proof of Theorem 1

Restate Theorem 1:

**Theorem 1.** *By optimizing Eqn (5), we maximize the mutual information between the view's embedding $Z_S$ and the input data $X$, and minimize the mutual information between the view's embedding $Z_S$ and the view it self $S$, conditioned on the input data $X$. Formally,*

$$\min_\theta \mathcal{L} \Rightarrow \max_\theta I(Z_S, X) \quad and \quad \min_\theta I(Z_S, S|X). \tag{33}$$

*Proof.* According to Remark 1, we have:

$$I(Z_S, S|X) = H(Z_S|X). \tag{34}$$

According to Property 5 in Appendix C, we have:

$$I(Z_S, X) = H(Z_S) - H(Z_S|X). \tag{35}$$

Then combining Proposition 1 and Proposition 2, we conclude the proof. $\qquad \square$

### D.4 Proof of Theorem 2

Restate Theorem 2:

**Theorem 2.** *Assume $0 < \beta \leq 1$, then by minimizing Eq. (5), the self-supervised Information Bottleneck objective is maximized, formally:*

$$\min_\theta \mathcal{L} \Rightarrow \max_\theta \mathcal{IB}_{ssl}. \tag{36}$$

*Proof.* According to Property 5 in Appendix C, we can rewrite the IB principle in SSL setting as:

$$\mathcal{IB}_{ssl} = [H(Z_S) - H(Z_S|X)] - \beta [H(Z_S) - H(Z_S|S)]. \tag{37}$$

Notice that $Z_S$ is deterministic given $S$: $Z_S = f_\theta(S)$. According to Property 5 in Appendix C, we have $H(Z_S|S) = 0$. Hence, we further have the following relationship

$$\mathcal{IB}_{ssl} = (1 - \beta)H(Z_S) - H(Z_S|X). \tag{38}$$

Let $\lambda = 1 - \beta \geq 0$. Now we can decompose the objective $\mathcal{IB}_{ssl}$ into two terms: 1) maximizing $H(Z_S)$, which increases the information entropy of the embeddings of augmented data. 2) minimizing $H(Z_S|X)$, which decreases the entropy of the embeddings of augmented data, conditioned on the original data.

With Proposition 1 and Proposition 2, we complete the proof. $\square$

### D.5 Proof of Corollary 1

Restate Corollary 1:

**Corollary 1.** *Let $X_1 = S$, $X_2 = X$ and assume $0 < \beta \leq 1$, then minimizing Eq. (5) is equivalent to minimizing the Multi-view Information Bottleneck Loss in [9]:*

$$\mathcal{L}_{MIB} = I(Z_1, X_1|X_2) - \beta I(X_2, Z_1), 0 < \beta \leq 1 \tag{39}$$

By maximizing $I(X_2, Z_1)$, the model could obtain sufficient information for downstream tasks by ensuring the representation $Z_1$ of $X_1$ is sufficient for $X_2$, and decreasing $I(Z_1, X_1|X_2)$ will increase the robustness of the representation by discarding irrelevant information.

*Proof.* Let $X_1 = S, X_2 = X$ be two views of the input data. We have:

$$
\begin{aligned}
\mathcal{L}_{MIB} &= I(Z_S, S|X) - \beta I(X, Z_S) \\
&= [H(Z_S|X) - H(Z_S|S, X)] - \beta[H(Z_S) - H(Z_S|X)] \\
&= (1 - \beta)H(Z_S|X) - \beta H(Z_S) - H(Z_S|S, X).
\end{aligned} \tag{40}
$$

As $Z_S$ is deterministic given $S$, we can obtain $H(Z_S|S, X) = 0$. Based on this, we can further simplify $\mathcal{L}_{MIB}$ as

$$\mathcal{L}_{MIB} = H(Z_S|X) - \lambda H(Z_S), \text{ with } \lambda > 0. \tag{41}$$

With Proposition 1 and Proposition 2, we complete the proof. $\square$

### D.6 Proof of Corollary 2

Restate Corollary 2:

**Corollary 2.** *When the data augmentation process is reversible, minimizing Eq. (5) is equivalent to learning the Minimal and Sufficient Representations for Self-supervision in [45]:*

$$Z_X^{ssl} = \underset{Z_X}{\arg\max}\, I(Z_X, S), Z_X^{ssl_{min}} = \underset{Z_X}{\arg\min}\, H(Z_X|S) \text{ s.t. } I(Z_X, S) \text{ is maximized.} \tag{42}$$

$Z_X^{ssl}$ is the sufficient self-supervised representation by maximizing $I(Z_X, S)$, and $Z_X^{\mathrm{ssl_{min}}}$ is the minimal and sufficient representation by minimizing $H(Z_X|S)$.

*Proof.* Eq. (42) can be converted to minimizing the relaxed Lagrangian objective as below

$$\mathcal{L}_{\mathrm{ssl_{min}}} = H(Z_X|S) - \beta I(Z_X, S), \text{ with } \beta > 0. \tag{43}$$

Then $\mathcal{L}_{\mathrm{ssl}_{min}}$ could be decomposed into

$$
\begin{aligned}
\mathcal{L}_{\mathrm{ssl_{min}}} &= H(Z_X|S) - \beta I(S, Z_X) \\
&= H(Z_X|S) - \beta[H(Z_X) - H(Z_X|S)] \\
&= (1 + \beta)H(Z_X|S) - \beta H(Z_X)
\end{aligned} \tag{44}
$$

With $\beta > 0$, $\mathcal{L}_{\mathrm{ssl_{min}}}$ is essentially a symmetric formulation of Eq. (38), by exchanging $X$ with $S$, and $Z_X$ with $Z_S$. With the assumption that the data augmentation process is reversible and according to Proposition 1 and Proposition 2, we conclude the proof. $\square$

### D.7 Proof of Theorem 3

Restate Theorem 3:

**Theorem 3** *(task-relevant/irrelevant information). By optimizing Eq. (5), the task-relevant informa-tion $I(Z_S, T)$ is maximized, and the task-irrelevant information $H(Z_S|T)$ is minimized. Formally:*

$$\min_\theta \mathcal{L} \Rightarrow \max_\theta I(Z_S, T) \text{ and } \min_\theta H(Z_S|T). \tag{45}$$

*Proof.* Note that with Assumption 2, we have $I(X, T|S) = I(S, T|X) = 0$. Therefore we obtain $0 \leq I(Z_S, T|X) \leq I(S, T|X) = 0$, which induces $I(Z_S, T|X) = 0$. Then we can derive

$$
\begin{aligned}
I(Z_S, T) =& I(Z_S, T|X) + I(Z_S, X, T) \\
=& 0 + I(Z_S, X) - I(Z_S, X|T) \\
=& I(Z_S, X) - I(Z_S, X|T) \\
\geq& I(Z_S, X) - I(X, S|T)
\end{aligned} \tag{46}
$$

and

$$
\begin{aligned}
H(Z_S|T) =& H(Z_S|X, T) + I(Z_S, X|T) \\
=& H(Z_S|X) - I(Z_S, T|X) + I(Z_S, X|T) \\
=& H(Z_S|X) - 0 + I(Z_S, X|T) \\
\leq& H(Z_S|X) + I(X, S|T)
\end{aligned} \tag{47}
$$

Note that $I(X, S|T)$ is a fixed gap indicating the amount of task-irrelevant information shared between $X$ and $S$.

With Theorem 1, by optimizing the objective Eq. (5), we maximize the lower bound of the task-relevant information $I(Z_S, T)$, and minimize the upper bound of the task-irrelevant information $H(Z_S|T)$. Then the proof is completed. □

## E  Implementation Details

### E.1  Loss function

In our implementation we did not directly use the original loss function as given in Eqn. (5). For simplicity, we use its equivalent form, which can be easily derived from the following equation:

$$
\begin{aligned}
\left\| \tilde{\mathbf{Z}}_A - \tilde{\mathbf{Z}}_B \right\|_F^2 =& \sum_{k=1}^{D} \sum_{i=1}^{N} (\tilde{z}_{i,k}^A - \tilde{z}_{i,k}^B)^2 \\
=& \sum_{k=1}^{D} \sum_{i=1}^{N} \left( (\tilde{z}_{i,k}^A)^2 + (\tilde{z}_{i,k}^B)^2 - 2 * \tilde{z}_{i,k}^A \tilde{z}_{i,k}^B \right) \\
=& \sum_{k=1}^{D} (2 - 2 * \tilde{Z}_{A,k}^\top \tilde{Z}_{B,k}) \\
=& 2D - 2 * \text{trace}(\tilde{\mathbf{Z}}_A^\top \tilde{\mathbf{Z}}_B)
\end{aligned} \tag{48}
$$

So we can rewrite the objective function Eqn. (5) as the following one:

$$\mathcal{L} = -\text{trace}(\tilde{\mathbf{Z}}_A^\top \tilde{\mathbf{Z}}_B) + \lambda' \left( \left\| \tilde{\mathbf{Z}}_A^\top \tilde{\mathbf{Z}}_A - \mathbf{I} \right\|_F^2 + \left\| \tilde{\mathbf{Z}}_B^\top \tilde{\mathbf{Z}}_B - \mathbf{I} \right\|_F^2 \right) \tag{49}$$

where the $\lambda'$ here should be half of the $\lambda$ in Eqn. (5). For simplicity we do not discriminate between these two symbols. The values of the trade-off parameter $\lambda$ in Fig 2 as well as that in Appendix E.4 are actually denoted as $\lambda'$ in Eqn. (49).

Table 6: Statistics of benchmark datasets

| Dataset | #Nodes | #Edges | #Classes | #Features |
|---|---|---|---|---|
| Cora | 2,708 | 10,556 | 7 | 1,433 |
| Citeseer | 3,327 | 9,228 | 6 | 3,703 |
| Pubmed | 19,717 | 88,651 | 3 | 500 |
| Coauthor CS | 18,333 | 327,576 | 15 | 6,805 |
| Coauthor Physics | 34,493 | 991,848 | 5 | 8,451 |
| Amazon Computer | 13,752 | 574,418 | 10 | 767 |
| Amazon Photo | 7,650 | 287,326 | 8 | 745 |

## E.2  Graph augmentations

We adopt two random data augmentations strategies on graphs: 1) **Edge dropping**. 2) **Node feature masking**. The two strategies are widely used in node-level contrastive learning [57, 58, 39].

- **Edge dropping** works on the graph structure level, where we randomly remove a portion of edges in the original graph. Formally, given the edge dropping ratio $p_e$, for each edge we have $p_e$ probability to drop this edge from the graph. When calculating the degree for each node, the dropped edge will not be considered.

- **Node feature masking** works on the node feature level, where we randomly set a fraction of features of all nodes as $0$. Formally, given the node feature masking ratio $p_f$, for each input feature, we set it as $0$ with a probability of $p_f$. Note that the masking operation is applied to the selected feature columns of all the nodes.

Note that the previous works [57, 58, 39] use two separate sets of edge dropping ratio $p_e$ and node feature dropping ratio $p_f$ for generating two views, i.e. $p_{e_1}$ and $p_{f_1}$ for view $A$, $p_{e_2}$ and $p_{f_2}$ for view $B$. However, in our implementation, we let $p_{e_1} = p_{e_2}$ and $p_{f_1} = p_{f_2}$, so that the two transformations $t_A$ and $t_B$ come from the same distribution $\mathcal{T}$.

## E.3  Datasets

We evaluate our models on seven node classification benchmarks: *Cora, Citeseer, Pubmed, Coauthor CS, Coauthor Physics, Amazon Computer* and *Amazon Photo*. We provide dataset statistics in Table 6, and brief introduction and settings are as follows:

**Cora[4], Citeseer, Pubmed[5]** are three widely used node classification benchmarks [34, 29]. Each dataset contains one citation network, where nodes mean papers and edges mean citation relationships. We use the public split for linear evaluation, where each class has fixed 20 nodes for training, another fixed 500 nodes and 1000 nodes are for validation/test respectively.

**Coauther CS, Coauther Physics** are co-authorship graphs based on the Microsoft Academic Graph from the KDD Cup 2016 challenge [35]. Nodes are authors, that are connected by an edge if they co-authored a paper; node features represent paper keywords for each author's papers, and class labels indicate most active fields of study for each author. As there is no public split for these datasets, we randomly split the nodes into train/validation/test (10%/10%/80%) sets.

**Amazon Computer, Amazon Photo** are segments of the Amazon co-purchase graph [26], where nodes represent goods, edges indicate that two goods are frequently bought together; node features are bag-of-words encoded product reviews, and class labels are given by the product category. We also use a 10%/10%/80% split for these two datasets.

For all datasets, we use the processed version provided by Deep Graph Library [49][6]. All datasets are public available and do not have licenses.

---

[4] https://relational.fit.cvut.cz/dataset/CORA
[5] Citeseer and Pubmed: https://linqs.soe.ucsc.edu/data
[6] https://docs.dgl.ai/en/0.6.x/api/python/dgl.data.html, Apache License 2.0

Table 7: Details of hyper-parameters of the experimental results in Table 2 and Table 3.

| Dataset | CCA-SSG | | | | | | | | Logistic Regression | |
|---|---|---|---|---|---|---|---|---|---|---|
| | Steps | # layers | # hidden units | $\lambda$ | lr | wd | $p_f$ | $p_e$ | lr | wd |
| Cora | 50 | 2 | 512-512 | 1e-3 | 1e-3 | 0 | 0.1 | 0.4 | 1e-2 | 1e-4 |
| Citeseer | 20 | 1 | 512 | 5e-4 | 1e-3 | 0 | 0.0 | 0.4 | 1e-2 | 1e-2 |
| Pubmed | 100 | 2 | 512-512 | 1e-3 | 1e-3 | 0 | 0.3 | 0.5 | 1e-2 | 1e-4 |
| Computer | 50 | 2 | 512-512 | 5e-4 | 1e-3 | 0 | 0.1 | 0.3 | 1e-2 | 1e-4 |
| Photo | 50 | 2 | 512-512 | 1e-3 | 1e-3 | 0 | 0.2 | 0.3 | 1e-2 | 1e-4 |
| CS[1] | 50 | 2 | 512-512 | 1e-3 | 1e-3 | 0 | 0.2 | - | 5e-3 | 1e-4 |
| Physics | 100 | 2 | 512-512 | 1e-3 | 1e-3 | 0 | 0.5 | 0.5 | 5e-3 | 1e-4 |

[1] We use MLP (instead of GCN) as the encoder on Coauthor-CS, which is essentially equivalent to setting $p_e = 1.0$ (drop all the edges except the self-loops).

In Table 2, we have mentioned that for MVGRL [15] and GRACE [57], we reproduce the experiments with authors' codes, both of which are publicly available: MVGRL[7] and GRACE[8].

### E.4 Hyper-parameters

We provide all the detailed hyper-parameters on the seven benchmarks in Table 7. All hyper-parameters are selected through small grid search, and the search space is provided as follows:

- Training steps: {20, 50, 100, 200}
- Number of layers: {1, 2, 3}
- Number of hidden units: {128, 256, 512, 1024}
- $\lambda$: {1e-4, 5e-4, 1e-3, 5e-3, 1e-2}
- learning rate of CCA-SSG: {5e-4, 1e-3, 5e-3}
- weight decay of CCA-SSG: {0, 1e-5, 1e-4, 1e-3}
- edge dropping ratio: {0, 0.1, 0.2, 0.3, 0.4, 0.5}
- node feature masking ratio: {0, 0.1, 0.2, 0.3, 0.4, 0.5}
- learning rate of logistic regression: {1e-3, 5e-3, 1e-2}
- weight decay of logistic regression: {1e-4, 1e-3, 1e-2}

## F   Additional Experiments

### F.1   Visualizations of Correlation Matrix

In Fig. 5, we provide visualizations of the absolute correlation matrix of the raw input features, the embeddings without decorrelation term, and embeddings with decorrelation term on three datasets: *Cora, Citeseer* and *Pubmed*.

As we can see, the raw input feature of the three datasets are all nearly fully uncorrelated (Fig. 5(a), 5(d) and 5(g)). Specifically, the on-diagonal term is close to 1 while the off-diagonal term is close to 0. When training without the decorrelation term Eq. (7), the off-diagonal elements of the correlation matrix of node embeddings increase dramatically as shown in Fig. 5(b) and 5(h), indicating that different dimensions fail to capture orthogonal information. Fig. 5(c) and 5(i) show that with the decorrelation term Eq. (7), our method could learn nearly highly disentangled representations. An interesting finding is that even without the decorrelation term, on Citeseer our method could still generate fairly uncorrelated representations (Fig. 5(e)). The possible reason is that: 1) on Citeseer, we use a one-layer GCN as the encoder, which is less expressive than a two-layer one and alleviate the trend of collapsing. 2) The number of training steps on Citeseer is much smaller than others, so that the impact of invariance term is weaken.

---

[7] https://github.com/kavehhassani/mvgrl, no license.
[8] https://github.com/CRIPAC-DIG/GRACE, Apache License 2.0.

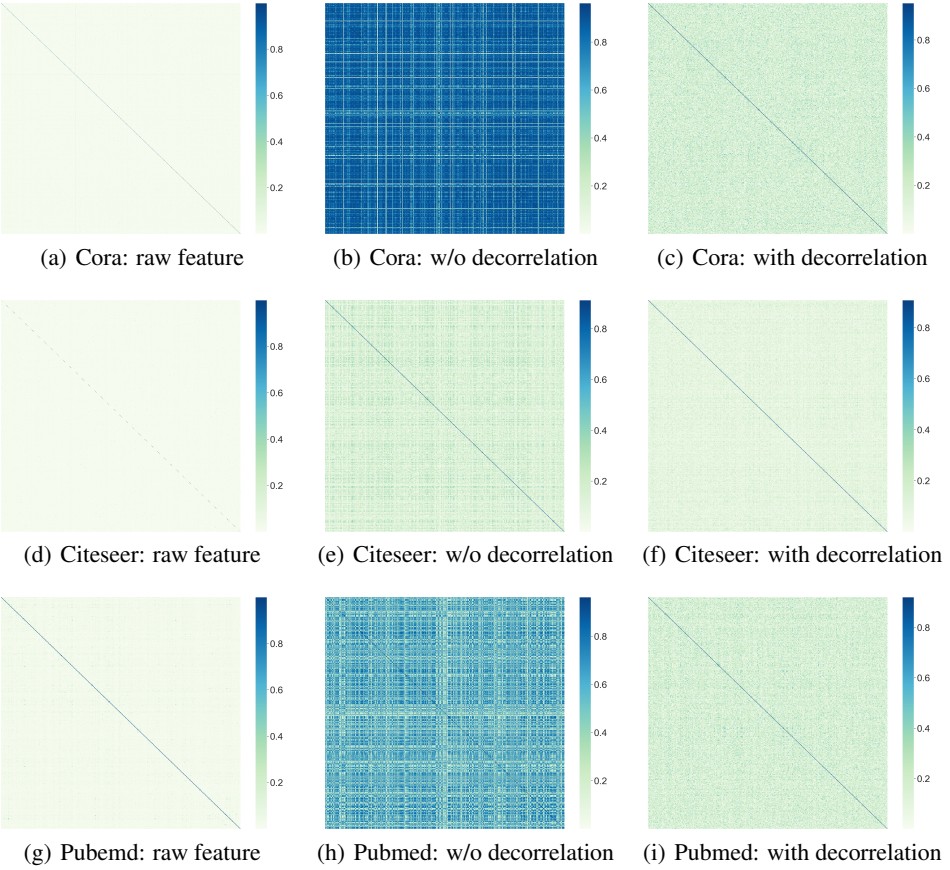

Figure 5: Visualizations of the correlation matrix (absolute value) of the raw input features, the embeddings without decorrelation term, and embeddings with decorrelation term on *Cora, Citeseer* and *Pubmed*. Light green: $\rightarrow 0$; Dark blue: $\rightarrow 1$.

These visualizations also echo the dimensional collapse issue as discussed in Appendix B: without the feature decorrelation term Eq. (7), there is a high probability that all the dimensions capture similar semantic information, thus leading to the dimensional collapse issue. The dimensional collapse can be fundamentally avoided by the decorrelation term Eq. (7).

## F.2 Effects of Augmentation Intensity

We further explore the effects of augmentation intensity on downstream node classification tasks. We try different combinations of the feature masking ratio $p_f$ and edge dropping ratio $p_e$, and report the corresponding performance on the 7 benchmarks mentioned in Appendix E.3. Other hyper-parameters are the same as reported in Table 7. As we can see in Fig. 6, for each dataset there exists an optimal $p_e/p_f$ combination, that could help the model reach the best performance. Also, we find that our method is not that sensitive to the augmentation intensity: as long as $p_e$ and $p_f$ are in a proper range, our method could still achieve impressive and competitive performance. However, it is still very important to select a proper augmentation intensity as well as augmentation method, in order for label-invariant data augmentations for learning informative representations.

## F.3 Performance under Low Label Rates

We further evaluate the node embeddings learned through CCA-SSG on downstream node classification tasks (still using linear, logistic regression), with respect to various label rates (ratio of training nodes). The experiments are conducted on three citation networks: *Cora, Citeseer* and *Pubmed*. In the linear evaluation step, we follow the setups in [24]: we train the linear classifier with 1%, 2%,

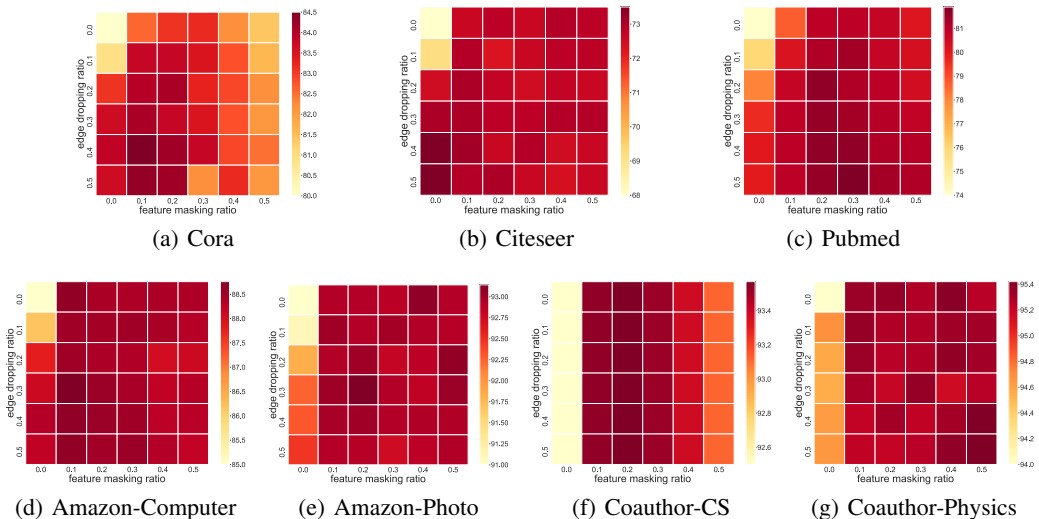

Figure 6: Visualizations of the effects of different augmentation intensity, by adopting different combinations of feature masking ratio $p_f$ and edge dropping ratio $p_e$, and we report test accuracy (%). Each row represents a specific setting for edge dropping ratio $p_e$ and each column represents a specific setting for feature masking ratio $p_f$. Note that when $p_e = p_f = 0$ (the upper left entry in each subfigure), the test accuracy for different datasets should be: 50.2 on *Cora*; 30.5 on *Citeseer*; 46.4 on *Pubmed*; 54.56 on *Computer*; 83.95 on *Photo*; 90.4 on *CS*; 87.85 on *Physics*. Since these results are much worse than the others, we raise their values in each subfigure for better visualization. On *CS*, the edge dropping ratio $p_e$ would make no difference to the performance as we use MLP as the encoder, which does not take graph structure as input.

Table 8: Node classification Accuracy under low label rates (%).

| Dataset | Cora | | | | | Citeseer | | | | | Pubmed | | |
|---|---|---|---|---|---|---|---|---|---|---|---|---|---|
| Label Rate | 1% | 2% | 3% | 4% | 5% | 1% | 2% | 3% | 4% | 5% | 0.05% | 0.1% | 0.3% |
| LP | 62.3 | 65.4 | 67.5 | 69.0 | 70.2 | 40.2 | 43.6 | 45.3 | 46.4 | 47.3 | 66.4 | 65.4 | 66.8 |
| Cheby | 52.0 | 62.4 | 70.8 | 74.1 | 77.6 | 42.8 | 59.9 | 66.2 | 68.3 | 69.3 | 47.3 | 51.2 | 72.8 |
| GCN | 62.3 | 72.2 | 76.5 | 78.4 | 79.7 | 55.3 | 64.9 | 67.5 | 68.7 | 69.6 | 57.5 | 65.9 | 77.8 |
| CCA-SSG | **72.5** | **79.3** | **81.0** | **82.0** | **82.3** | **58.9** | **65.6** | **68.6** | **70.8** | **71.7** | **68.8** | **73.1** | **81.1** |

3%, 4%, 5% (resp. 0.05%, 0.1%, 0.3%) training nodes on *Cora* and *Citeseer* (resp. *Pubmed*), and then test the model with another 1000 nodes. Both training nodes and testing nodes are randomly selected for each trial, and we report the mean accuracy through 20 trials with random splits and random initialization in Table 8.

We compare our method with Label Propagation, GCN with Chebyshev filter(Cheby) and the vanilla GCN [22], whose results are taken from [24] as well. As we can see in Table 8, our method achieves very impressive performance under low label rates, especially when the labeled nodes are really scarce (i.e. 1% on *Cora* and *Citeseer*, 0.05% on *Pubmed*). This is because through self-supervised pretraining, our method could fully utilize the information of unlabeled nodes, and learn good representations for them, which make them easy to distinguish even with only a few number of labeled nodes for training.

# G   Further Comparisons with previous contrastive methods

In Table 1 we have made a thorough comparison with typical contrastive methods from the **technical details**. Here, we further compare our method with more existing contrastive self-supervised graph models (both node-level and graph-level) from the perspective of their general, conceptual designs: 1) How they generate views. 2) The pairs for contrasting. 3) The loss function. 4) Downstream tasks

Table 9: Further conceptual comparison with existing contrastive learning methods on graphs. *View generation in general* (how the method generate views): Cross-scale means this method treat elements in different scales of the graph data as different views (e.g. node and graph); Fix-Diff means using fixed graph diffusion [23] operation to create another view; Rand-Aug means using random graph augmentations (e.g. edge dropping, feature masking, etc.) to generate views. *Pairs* represents the contrasting components, where $N$ is node and $G$ is graph. *Loss* (i.e. the used loss function): NCE represents Noise-Contrastive Estimation [13]; JSD represents Jensen-Shannon Mutual Information Estimator [30]; InfoNCE represents InfoNCE Estimator [46]; MINE means Mutual Information Neural Estimator [3]; BYOL means the asymmetric objective proposed in the BYOL paper [12]. *Tasks* denotes the downstream tasks (node-level, graph-level or edge-level) to which the method has been applied.

| | Methods | View generation in general | Pairs | Loss | Tasks |
|---|---|---|---|---|---|
| Instance-level | DGI [48] | Cross-scale | N-G | NCE | Node |
| | InfoGraph [38] | Cross-scale | N-G | JSD | Graph |
| | MVGRL [15] | Fix-Diff + Cross-scale | N-G | NCE/JSD | Node/Graph |
| | GCC [33] | Rand-Aug | N-N | InfoNCE | Node/Graph |
| | GMI [31] | Hybrid[1] | Hybrid | MINE/JSD | Node/Edge |
| | GRACE [57] | Rand-Aug | N-N | InfoNCE | Node |
| | GraphCL [52] | Rand-Aug | G-G | InfoNCE | Graph |
| | GCA [58] | Rand-Aug | N-N | InfoNCE | Node |
| | CSSL [54] | Rand-Aug | G-G | InfoNCE | Graph |
| | IGSD [55] | Rand-Aug | G-G | BYOL+InfoNCE | Graph |
| | GraphLog [28] | Rand-Aug | N-P-G[2] | InfoNCE | Graph |
| | BGRL [39] | Rand-Aug | N-N | BYOL | Node |
| | MERIT [27] | Fix-Diff + Rand-Aug | N-N | BYOL+InfoNCE | Node |
| | CCA-SSG (Ours) | Rand-Aug | F-F | CCA | Node |

[1] The view generation and contrasting pairs in GMI [31] is unique and complex, and could not be classified into any category.

[2] P denotes hierarchical prototype and could be seen as clustering centroid.

(i.e., node-level, edge-level or graph-level). The comparison is shown in Table 9. Note that this is a high-level comparison with general taxonomy, and each method may have distinct implementation details and specific designs.

We highlight that all of the previous methods focus on contrastive learning at instance level. Our paper proposes a non-contrastive and non-discriminative objective as a new self-supervised representation learning framework, inspired by canonical correlation analysis.