# OpenReview forum: "From Canonical Correlation Analysis to Self-supervised Graph Neural Networks"
_NeurIPS.cc/2021/Conference — NeurIPS 2021 Poster_

### Official Review · Reviewer_nj2q · 2021-07-16

**Rating:** 7
**Confidence:** 4

**Summary:**

This paper focuses on self-supervised graph neural networks, aiming at training graph neural networks without labeled data, and an approach called the CCA-SSG is proposed. CCA-SSG constructs two views of the given graph through edge dropping and node feature masking, and further uses CCA for learning node representations from the two views. The idea of formalizing unsupervised node representation learning as a multi-view learning task looks novel to me. Also, CCA-SSG has intuitive connections with the information bottleneck principle, which is attractive. The authors do experiment on multiple datasets, and CCA-SSG outperforms many competitive baseline methods.

Strengths:
1. Important problem.
2. Principled method with good theoretical connections.
3. Strong results.

**Main Review:**

Below are some detailed comments and questions:

1. At line 553 of the appendix, it is said that s_1 and s_2 come from the same distribution p_{aug}(x). But from my understanding of the proposed method, s_1 and s_2 are graphs from two different views (i.e., s_1 from edge dropping and s_2 from node feature masking respectively). If this is the case, s_1 and s_2 should come from different distributions. Is it correct?

2. CCA-SSG relies on edge dropping and node feature masking to augment the original graph, but the hyperparameters of edge dropping rate and feature masking rate are not studied in the paper. How will the results of CCA-SSG vary if the hyperparameters are changed?

**Time Spent Reviewing:**

3 hours

---

> ### Author Response · Authors · 2021-08-06
> **Response to the raised questions**
>
> Thanks for the nice review of the paper. The main questions concentrate on the data augmentation process and related ablation experiments.
>
> Q1: The data augmentation process
>
> $s_1$ and $s_2$ are indeed graphs from two different views; however, they are i.i.d samples from the same augmentation distribution, which is determined by the edge dropping rate $p_e$ and feature masking rate $p_f$. Note that we use both edge dropping and feature masking for both views (not edge dropping only for $s_1$, and feature masking only for $s_2$ ), and with identical rates, i.e. we use $p_e = 0.2$ and $p_f = 0.3$ to generate two views at the same time. This principle is discussed further in Appendix E.2, from line $657$ to line $660$, and the hyper-parameter details in Table 7 strictly follow it.
>
> Q2. Experiments on augmentations with varying hyperparameters
>
> Actually we did this experiment in Appendix F.2, where we explore the effects of augmentation intensity on downstream node classification tasks. We try different combinations of the feature masking ratio $p_f$ and edge dropping ratio $p_e$, and report the corresponding performance on the 7 benchmarks in Fig.6. Through this experiment, we find that 1) although there exist optimal $p_e$ / $p_f$ combinations for each dataset, our method is not that sensitive to the augmentation intensity. 2) It is still very important to select a proper augmentation intensity as well as augmentation method, in order for label-invariant data augmentations for learning informative representations.

---

> > ### Comment · Reviewer_nj2q · 2021-08-31
> > **Response to the Authors**
> >
> > Thanks the authors for the response, which addresses most of my concerns. Overall, I think this is a good paper, and I will keep my score unchanged.

---

### Official Review · Reviewer_N2Hb · 2021-07-17

**Rating:** 6
**Confidence:** 3

**Summary:**

The authors propose CCA-SSG for self-supervised representation learning on graph. Tee method is simple and efficient. Experimental results show high performance of the method on the node classification task.


**Limitations And Societal Impact:**

The details of the proposed method in this paper are not clear.

The method is unable to extract features involving the minority.


**Main Review:**

The problem setting is not clear.
The basic objective of CCA is to extract two different correlated features from two different observed data sets.
The link between CCA and node representation is not clear.
There is almost no explanation on the details of datasets in the experimental section.
Each dataset has two different observations?

The authors made a comparison of the performance of many methods, but traditional CCA is missing. Traditional CCA can not be applied to the node representation learning problem? The title of this paper is "From Canonical Correlation Analysis to Self-supervised Graph Neural Networks". It would be better to show the performance of CCA as well.

For the label prediction, why did the authors use logistic regression?

From Table 2, unsupervised methods tend to work better than supervised methods. It is not strange?

The perfromance is heavily dependent on lambda and the embedding dimension. How can appropriate parameters be determined in practice?



Thank you for the answers.
I have raised my score.

**Time Spent Reviewing:**

4

---

> ### Author Response · Authors · 2021-08-06
> **Thorough discussions for addressing the concerns one-by-one**
>
> We appreciate the thorough reviews. Below we address each point, and try to resolve several issues that are critical to understanding our contribution.
>
> Q1. Problem setting is unclear.
>
> We provide an elaboration which we hope can help to resolve the big picture. Our focus is on learning self-supervised node classification models, a well-established problem that has received significant attention in recent years [13, 32, 41, 48, 49]. The model input is a graph $G$, with node features $X$ and graph structure (adjacency matrix) $A$. The goal is then to learn, without labeled training data, node representations that can later be applied to downstream tasks. The key point of self-supervised learning is that, while we do not have downstream task information such as node labels, we can nonetheless design some signals as 'pseudo supervision' that can be useful for learning a good representation. Such a setting and learning paradigm is of much practical significance since in many cases labeling data can be costly. The key challenge is how to design effective self-supervision for representation learning without labels.
>
> Q2. What is the connection between CCA and our representation learning problem.
>
> While vanilla CCA has been proposed to extract two different correlated features from two different observed data sets (two random variables), its extended versions like Deep CCA [2] and Soft CCA [3] adopt CCA to learn low-dimensional representations of multi-view data. In our work, we identify that CCA provides a simple yet effective way for learning representations in an unsupervised way, and can be developed to solve self-supervised learning (on graphs) from a brand new perspective. Our key insight is that the objective induced by CCA in the context of self-supervised learning is sufficient for learning desirable representations.
> First, the representations can learn to preserve the shared information between two views (through the first term of our objective function), which provides an effective form of self-supervision. Second, the representation of each single view can be disentangled (through the second term of our objective function), which serves as a necessary regularization. This observation is backed up with our theoretical analysis and extensive empirical evidence.
> In our model, we create two views of the graph through random data augmentation (i.e., edge dropping and node feature masking), so the idea of CCA can be naturally adopted for self-supervised node representation learning.
>
>
> Q3. Need more details regarding datasets.
>
> Unfortunately with limited space, we had to defer some dataset details to the supplementary.  For example, in Appendix E.2 (which is referred at line 252 in the main text), we provide detailed information for all the datasets. In the experiments we use 7 datasets, including 3 citation networks Cora, Citeseer, Pubmed and 4 social networks Amazon-Computer, Amazon-Photo, Coauthor-CS and Coauthor-Physics. They are commonly used benchmarks for self-supervised node classification [32, 48, 49,]. For each dataset, as is illustrated in line 112-120, we create two views for each node through random augmentation (i.e., edge dropping and node feature masking). Such an operation is a common practice done by other self-supervised approaches for graph data [32, 48, 49].
>
> Q4. Results of traditional CCA are missing.
>
> This is a good suggestion. We provide new experiments using a traditional linear CCA model for self-supervised learning. The test accuracy results are: Cora $0.589 \pm 0.015 $, Citeseer $0.275 \pm 0.013$, Pubmed $0.758 \pm 0.004$, Amazon-Computer $ 0.7984 \pm 0.0053$, Amazon-Photo $ 0.8692 \pm 0.0072$, Coauthor-CS $0.9313 \pm 0.0018$, Coauthor-Physics $0.9504 \pm 0.0017$. Note that as the vanilla CCA uses a linear transformation as encoder, and does not consider the graph structure at all, it performs quite poorly on most datasets (e.g., Cora). However, when the graph structures are not that necessary for classification, the vanilla CCA can still yield decent performance (e.g. Coauthor-CS). We will add these results in the final version, which will further strengthen our contribution.
>
> Q5. Why is logistic regression used for label prediction.
>
> Unsupervised pertaining + linear model evaluation is a standard paradigm for the evaluation of unsupervised methods. In our model, we are required to evaluate the node embeddings learned from the unsupervised model, so we need to use an additional linear model to evaluate the embeddings on downstream tasks (e.g. node classification). In fact, any linear model, such as multi-class logistic regression and linear SVM, could be adopted for evaluation. As the majority of the competitive methods [13, 32, 41, 48, 49] use logistic regression, we also use it for fair comparison with them.
>
> Q6. How can unsupervised methods tend to work better than supervised methods.
>
> Although possibly counterintuitive at first glance, it is actually common for unsupervised methods to work better in the context of (semi-supervised) node classification tasks. Note that in this context, only a few number of nodes are included in the training set (e.g. on Pubmed, only 60 out of 19717 nodes are used as training nodes), which means most of the nodes are downplayed by the supervised model. However, in self-supervised methods (including ours), all the nodes will contribute equally during the unsupervised training process. The learned node representations will then be easier to be separated by the linear model for downstream tasks. So it is not strange that our method would outperform the supervised one. Also, the empirical evidence from previous works [13, 41, 48, 49] also indicates that unsupervised models can at times outperform their supervised competitors.
>
> Q7. Choosing $\lambda$ and the embedding dimension $D$.
>
> From previous works [13, 41, 48, 49], we observe that $256$ or $512$ are appropriate dimensions for nearly all datasets, whatever the size of the dataset (i.e., number of nodes within the dataset). So we just initialized the embedding dim as $512$, and tried other dims using grid search. Throughout our experiments we obtain the same observation that the embedding dimension of $512$ is optimal for the seven datasets we selected. The hyper-parameter $\lambda$ controls the trade-off between the invariance term and the decorrelation term. Note that if the embedding dimension is $D$, then the invariance term has $D$ entries, while the decorrelation term has $D^2$ entries. So intuitively $\lambda$ should be $1/D$ to balance the two terms. In our experiments, when $D$ is set as $512$, i.e., $1/D ~= 0.002$, we find that the optimal $\lambda$ would be $0.001$ for most datasets, which is very close to our hypothesis. When applying our method in practice, we recommend that $D$ and $\lambda$ be set as $512$ and $0.001$ as initial states, which could be then fine-tuned for different datasets and tasks.
>
> Q8. Some details of the proposed method in this paper are not clear.
>
> Because of space limitations, some details of our method were deferred to the supplementary.  In this regard, we have put all the details of the model, including the data augmentations, the loss function, the datasets, and the detailed hyper-parameters in Appendix E. This leaves more space for the theoretical foundations of our method (in Sec 4), which is arguably more important, as the proposed model is a very general method. Also, we provided the code for all the datasets in the supplementary materials.
>
> We hope these responses help to alleviate concerns, and look forward to receiving feedback regarding any remaining questions at the reviewer's earliest convenience.
>
>
> **References:**
>
> [2] Galen Andrew, Raman Arora, Jeff A. Bilmes, and Karen Livescu. Deep canonical correlation analysis. In ICML, pages 1247–1255, 2013.
>
> [3] Xiaobin Chang, Tao Xiang, and Timothy M. Hospedales. Scalable and effective deep CCA via soft decorrelation. In CVPR, pages 1488–1497, 2018.
>
> [13] Kaveh Hassani and Amir Hosein Khas Ahmadi. Contrastive multi-view representation learning on graphs. In ICML pages 4116–4126, 2020.
>
> [32] Shantanu Thakoor, Corentin Tallec, Mohammad Gheshlaghi Azar, Rémi Munos, Petar Velickovic,  and Michal Valko.  Bootstrapped representation learning on graphs. arXiv preprint arXiv:2102.06514, 2021.
>
> [41] Petar Velickovic, William Fedus, William L. Hamilton, Pietro Liò, Yoshua Bengio, and R. Devon Hjelm. Deep graph infomax. In ICLR, 2019.
>
> [48] Yanqiao Zhu, Yichen Xu, Feng Yu, Qiang Liu, Shu Wu, and Liang Wang. Deep graph contrastive representation learning.arXiv preprint arXiv:2006.04131, 2020.
>
> [49] Yanqiao Zhu, Yichen Xu, Feng Yu, Qiang Liu, Shu Wu, and Liang Wang. Graph contrastive learning with adaptive augmentation. In WWW, 2021.

---

> ### Author Response · Authors · 2021-08-22
> **Restatement of our contribution that we hope can help for your re-considering the assessment**
>
> Dear reviewer N2Hb, we sincerely hope our posted response can help for addressing the lingering points of concern. Below we provide a short-version summary of our contributions:
>
> - Our main contribution lies in the CCA-based objective function for self-supervised representation learning. Different from the previous InfoNCE-based objective function, ours does not require negative examples, which could reduce both the time and memory complexity for self-supervised learning.
>
> - The second contribution is that we provide theoretical guarantees for the proposed objective function based on Information Theory. We show that our objective function is an instantiation of the well-established Information Bottleneck Principle under self-supervised settings. Furthermore, we show that the learned representations would maximize the task-relevant information and minimize the task-irrelevant information, which provides theoretical justification for why our method would learn meaningful representations for downstream tasks.
>
> - Based on the proposed objective function, we design a simple yet effective pipeline for self-supervised node representation learning -- CCA-SSG. Our method is much simpler and time/memory efficient than the previous models, but yields competitive and superior performance on seven common benchmarks.
>
> We notice that your concerns mainly lie in the third contribution (the model design), which we hope has been addressed in our previous response. However, the first two contributions, which are more significant, might be neglected.  We hope this message could help to better elaborate our contributions and look forward to your follow-up comments.

---

> ### Author Response · Authors · 2021-08-31
> **Ask for re-assessment of our work**
>
> Regarding the initial review from reviewer N2Hb, we just want to reiterate that there are very clear-cut answers to every question/issue that was raised, and our rebuttal has carefully addressed each point-by-point.  Given then that these issues have all been resolved, combined with the positive feedback from other reviewers, we sincerely hope that the reviewer can reconsider the rating.

---

> > ### Comment · Reviewer_N2Hb · 2021-09-06
> > **re-assessment**
> >
> > Thank you for the answers to my questions. My concerns have been partially addressed. I have raised my score.
> > p.s.
> > I'm sorry that my response is late. I was sick in bed.

---

### Official Review · Reviewer_1tQC · 2021-07-17

**Rating:** 6
**Confidence:** 4

**Summary:**

This work proposes an alternative self-supervised objective inspired by canonical correlation analysis. Theoretical proof shows its connection to mutual information (MI) estimator and information bottleneck principles. Empirical experimental results demonstrate the effectiveness on node classification.

**Limitations And Societal Impact:**

I admit that the limitations and societal impact are given by authors.

**Main Review:**

The proposed self-supervised objective shares the same idea on making the representations of augmented views from the same data be close as much as possible. While the difference lies at the usage of negative samples. The proposed loss function does not need negative sample, but a regularization term instead. The key idea of this work is very easy to follow, and the experimental results on node classification task demonstrates its superiority to baselines.

Concerns:
I have a question about the correctness of the theoretical proof to show its connection to previous approach needs more explanation. It seems to be very important to dissect the proposed loss function. Authors drop an assumption that S denotes the augmented view from original input data X, and they share the same space, referring to Section 4, line 185. Then two different distributions (i.e., P(Z_S) and P(Z_S | X)) for the variable Z_S are defined in Line 191. Theoretical analysis are conducted based on this assumption. However, I'm confused about the difference between P(Z_S) and P(Z_S | X). It's easy to understand the variable Z_S depends on X because the augmented view S is generated from X. But it's difficult to understand why we can define a marginal distribution P(Z_S). It's not reasonable because S still depends on X. It causes a big trouble to follow the conclusion in Proposition 2. Why we can discuss the entropy of variable Z_S without the condition on input data X.

Minor Issue:
1. In the appendix D, please double check the Remark 1. How can we get the equality H(Z_S|X) - H(Z_S | S, X) = H(Z_S | X) according to property 6?

**Time Spent Reviewing:**

5

---

> ### Author Response · Authors · 2021-08-06
> **Detailed elaborations for resolving questions**
>
> Thanks for the careful comments. The reviewer appreciates our simple and effective approach as well as our experimental results. The main concern lies in the difference between the conditional distribution $P(Z_S | X)$ and the marginal distribution $P(Z_S)$ in our model, which the reviewer found confusing. Below we provide a detailed elaboration to address this concern.
>
> For convenience, we first re-express the notations in our paper. $X$ is a random variable representing the input data, $S$ is the corresponding random variable of the augmentation of the input data, and $Z_S$ is the random variable of the low-dimensional representation of $S$. So we have the following Markov Chain: $X\rightarrow S \rightarrow Z_S \ $. We use lower-case letters $x, s$ and $z_s$ to denote their corresponding realizations, i.e. the input data point, the data point's augmentation, and the subsequent low-dimensional representation. Since $Z_S$ is just a deterministic function of $S$ as instantiated via a GNN model (i.e., there is no randomness in $Z_S$ when conditioned on $S$), we remove $S$ from the Markov Chain and simplify it as $X \rightarrow Z_S$. Thus we have the following joint distribution of this directional graph: $P(Z_S, X) = P(X) \cdot P(Z_S | X)$.
>
> Based on the above facts, we have $P(Z_S) = \int P(Z_S, X) \mathrm{d}X = \int P(Z_S|X) P(X)  \mathrm{d}X $. This equation gives the relationship between $P(Z_S)$ and $P(Z_S|X)$. Essentially, $P(Z_S|X)$ is a function of $X$: when $X$ has different values, $P(Z_S|X)$ could be different and the dependence on $X$ cannot be ignored. In contrast, $P(Z_S)$ removes the effect of $X$ through integration (i.e., marginalization of the joint distribution).
>
> Let us go into a more concrete example for further illustration. Assume that $x_1, x_2, \cdots x_n$ are data points of the dataset, $s_1, s_2, \cdots, s_n$ are the corresponding augmentations, and $z_{s_1}, z_{s_2}, \cdots, z_{s_n}$ are the corresponding representations. Then we say $x_1, x_2, \cdots x_n$ are $n$ i.i.d samples from $P(X)$, and $z_{s_1}, z_{s_2}, \cdots, z_{s_n}$ are $n$ i.i.d samples from $P(Z_S)$. How about $P(Z_S|X)$? Suppose that given $x_1$, we generate $k$ augmentations whose representations are $z^1_{s_1}, z^2_{s_1}, \cdots, z^k_{s_1}$. Then we say  $z^1_{s_1}, z^2_{s_1}, \cdots, z^k_{s_1}$ are $k$ i.i.d. samples from $P(Z_S|x_1)$.
>
>
> **Minor question**: How can we get $H(Z_S|X) - H(Z_S | S, X) = H(Z_S | X)$.
>
> This is a good question. Since $ 0 \le H(Z_S|S, X) = H(Z_S|S) - I(Z_S,X|S) \le H(Z_S|S) = 0$, we have $H(Z_S|S,X)$ = 0. Therefore we have $H(Z_S|X) - H(Z_S | S, X) = H(Z_S | X)$. We will add this detailed derivation in our paper to make the presentation easier to follow.
>
> We hope the above elaborations will help to resolve any lingering points of confusion. And if there are any further questions, please let us know as soon as possible.

---

> ### Author Response · Authors · 2021-08-22
> **Thank you for the time and we hope our response helps to address your questions.**
>
> Dear reviewer 1tQC, we sincerely hope our posted response can help to address your question about the differences between $P(Z_S)$ and $P(Z_S | X)$  and serve as a reference for your re-assessment of our work. If you have any further comments and questions, please let us know and we are glad to write a follow-up response.

---

> > ### Comment · Reviewer_1tQC · 2021-08-30
> > **After reading the response**
> >
> > Thanks for the authors' reply. My concerns have been addressed well. I'd like to raise my rating to 6. But I still suggest authors to spend more time on checking the mathematical derivation to make it as clear as possible.

---

### Decision · Program_Chairs · 2021-09-27

**Decision:**

Accept (Poster)

**Comment:**

This paper proposes a new objective and framework for self-supervised representation learning on graphs. It formalizes unsupervised node representation learning as a multi-view learning task. Results show that the proposed approach outperforms many competitive baseline methods. The idea presented in the paper is interesting. The proposed approach depends on several hyperparameters. The effect of these hyperparameters are not studied and should be included.